EMBO
Molecular Medicine

# Liver ASK1 protects from non-alcoholic fatty liver disease and fibrosis

Tenagne D Challa[1,2], Stephan Wueest[1,2], Fabrizio C Lucchini[1,2,3], Mara Dedual[1,2,3], Salvatore Modica[4], Marcela Borsigova[1,2], Christian Wolfrum[4], Matthias Blüher[5] & Daniel Konrad[1,2,3,*]

## Abstract

**Non-alcoholic fatty liver disease (NAFLD) is strongly associated with obesity and may progress to non-alcoholic steatohepatitis (NASH) and liver fibrosis. The deficit of pharmacological therapies for the latter mainly results from an incomplete understanding of involved pathological mechanisms. Herein, we identify apoptosis signal-regulating kinase 1 (ASK1) as a suppressor of NASH and fibrosis formation. High-fat diet-fed and aged chow-fed liver-specific ASK1-knockout mice develop a higher degree of hepatic steatosis, inflammation, and fibrosis compared to controls. In addition, pharmacological inhibition of ASK1 increased hepatic lipid accumulation in wild-type mice. In line, liver-specific ASK1 overexpression protected mice from the development of high-fat diet-induced hepatic steatosis and carbon tetrachloride-induced fibrosis. Mechanistically, ASK1 depletion blunts autophagy, thereby enhancing lipid droplet accumulation and liver fibrosis. In human livers of lean and obese subjects, ASK1 expression correlated negatively with liver fat content and NASH scores, but positively with markers for autophagy. Taken together, ASK1 may be a novel therapeutic target to tackle NAFLD and liver fibrosis.**

**Keywords** autophagy; high-fat diet; NASH
**Subject Categories** Digestive System; Immunology; Metabolism
**EMBO Mol Med (2019) e10124**

## Introduction

Non-alcoholic fatty liver disease (NAFLD) is the most common chronic liver disease affecting up to 40% of adults and children (Imajo *et al*, 2012; Koyama & Brenner, 2017; Friedman *et al*, 2018). It has become a major public health problem in developed countries (Zhang *et al*, 2016b; Koyama & Brenner, 2017). The spectrum of NAFLD ranges from simple hepatic lipid accumulation (hepatic steatosis) to non-alcoholic steatohepatitis (NASH), which can

progress to liver fibrosis, cirrhosis, and hepatocellular carcinoma (Imajo *et al*, 2012; Meex *et al*, 2015; Zhang *et al*, 2016b). NAFLD is strongly associated with obesity and obesity-related metabolic disorders such as glucose intolerance, type 2 diabetes (T2D), and dyslipidemia (Sun *et al*, 2012; Zhang *et al*, 2016b). Increased uptake of free fatty acids (FFA), elevated *de novo* lipogenesis, decreased fat oxidation, reduced hepatic very low-density lipoprotein (VLDL) secretion, and impaired autophagy may contribute to the development of hepatic steatosis (Kohjima *et al*, 2007; Benhamed *et al*, 2012; Hur *et al*, 2016). The latter is induced during nutrient deprivation or stress conditions to maintain cellular homeostasis (Wang, 2015). Besides sequestrating damaged organelles, protein aggregates, and intracellular pathogens (McEwan & Dikic, 2011; Tang *et al*, 2012; Wang & Qin, 2013), autophagy degrades lipids, glycogen, and proteins to provide fuel and maintain energy homeostasis (Czaja, 2011; Gonzalez-Rodriguez *et al*, 2014; Inokuchi-Shimizu *et al*, 2014). Of note, activation of autophagy promotes anti-inflammatory pathways (Lodder *et al*, 2015), and impaired autophagy induces steatosis and liver fibrosis (Yang *et al*, 2010; Gonzalez-Rodriguez *et al*, 2014; Inokuchi-Shimizu *et al*, 2014; Stienstra *et al*, 2014; Lodder *et al*, 2015).

Fibrosis is the most important predictor of liver-related complications in NAFLD (Loomba *et al*, 2017). It is the result of sustained wound-healing processes in response to chronic hepatocyte injury leading to inflammatory reaction as well as increased production of extracellular matrix (ECM) proteins (Berres *et al*, 2010). Although initial triggering factors in the development of liver fibrosis remain largely unknown, the positive effects of autophagy in the prevention of hepatic steatosis and fibrosis (Singh *et al*, 2009; Yang *et al*, 2010; Czaja, 2011; Ding *et al*, 2014; Gonzalez-Rodriguez *et al*, 2014; Inokuchi-Shimizu *et al*, 2014; Lodder *et al*, 2015) have uncovered therapeutic benefit of pathways regulating autophagy to tackle NAFLD and liver fibrosis. Since deletion of autophagy genes may cause neonatal lethality (Takamura *et al*, 2011), it may be beneficial to modulate upstream signaling cascades that regulate autophagy.

Apoptosis signal-regulating kinase 1 (ASK1) is a member of the mitogen-activated protein kinase kinase kinase (MAP3Ks) family

1 Division of Pediatric Endocrinology and Diabetology, University Children's Hospital, Zurich, Switzerland
2 Children's Research Center, University Children's Hospital, Zurich, Switzerland
3 Zurich Center for Integrative Human Physiology, University of Zurich, Zurich, Switzerland
4 Institute of Food, Nutrition and Health, ETH Zurich, Schwerzenbach, Switzerland
5 Department of Medicine, University of Leipzig, Leipzig, Germany
*Corresponding author. Tel: ++41-44-266 7966; Fax: ++41-44-266 7983; E-mail: daniel.konrad@kispi.uzh.ch

and an upstream activator of the c-Jun N-terminal kinase (JNK) and p38 MAPK signaling cascades (Ichijo *et al*, 1997; Ichijo, 1999; Hsieh & Papaconstantinou, 2006; Imoto *et al*, 2006). ASK1 is activated by various stressors such as reactive oxygen species (ROS), tumor necrosis factor alpha (TNF-α), endoplasmic reticulum (ER) stress, and lipopolysaccharides (LPS; Imoto *et al*, 2006; Rudich *et al*, 2007; Saito *et al*, 2007; Tobiume *et al*, 2002), placing ASK1 as a signaling node in which different stressors converge. These stress signals phosphorylate ASK1 to subsequently induce activation of JNK and p38 MAPK (Pan *et al*, 2010; Nakagawa *et al*, 2011). In turn, activated JNK and p38 MAPK regulate autophagy, thereby modulating apoptosis, proliferation, and inflammation to maintain cellular integrity (Ichijo *et al*, 1997; Tobiume *et al*, 2001; McEwan & Dikic, 2011; Nakagawa *et al*, 2011; Sui *et al*, 2014). In support of an important role of ASK1 in the regulation of autophagy, whole body ASK1-knockout mice are highly susceptible to lethal bacterial infection due to blockage of autophagy in the liver (Gade *et al*, 2014). Herein, we hypothesized that liver-specific ASK1 plays a crucial role in the regulation of hepatocyte autophagy and, consequently, in the development of NAFLD and fibrosis.

# Results

### Decreased liver ASK1 expression in human subjects with hepatic steatosis and NASH

To assess whether ASK1 may contribute to the development of hepatic steatosis and NASH in humans, we analyzed *ASK1* mRNA expression in the liver of lean and obese subjects with or without type 2 diabetes. Basic clinical characteristics of these subjects are provided in Appendix Table S1. We found significant negative correlations between liver *ASK1* expression and BMI, plasma alanine aminotransferase (ALAT), and liver fat content (Fig 1A–C). *ASK1* expression was significantly reduced in patients with elevated NASH scores (Fig 1D). Moreover, *ASK1* expression correlated positively with the autophagy marker *autophagy-related gene 5 (ATG5), 7 (ATG7),* and *12 (ATG12)* expression (Fig 1E–G). These data suggest a protective role for liver-expressed *ASK1* in the development of hepatic steatosis as well as NASH in humans, potentially via regulating autophagy.

### Depletion of hepatic ASK1 increases lipid storage

To elucidate a potential role of ASK1 in hepatic lipid accumulation, experiments in cultured hepatocytes were performed. As shown in Appendix Fig S1A, ASK1 protein is expressed in HepG2 cells and treatment with palmitate and/or rapamycin increased its phosphorylation. The latter is a well-known activator of ASK1 and autophagy (Huang *et al*, 2004; Gonzalez-Rodriguez *et al*, 2014). As expected, rapamycin significantly decreased palmitate-induced lipid droplet accumulation in hepatocytes (Osawa *et al*, 2011; Fig 2A). To assess the involvement of ASK1 in rapamycin-induced lipid clearance, ASK1 was depleted in HepG2 cells using siRNA targeting ASK1 (siASK1; Fig 2B and Appendix Fig S1B). As expected, ASK1 silencing decreased phosphorylation of its downstream target JNK in hepatocytes treated with BSA or palmitate (Appendix Fig S1C).

Compared to cells transfected with non-targeting control siRNA (siCtrl), ASK1 depletion significantly increased lipid droplet accumulation in hepatocytes treated with or without palmitate (Fig 2C; Appendix Fig S1D). Importantly, the effect of rapamycin on the clearance of palmitate-induced lipid accumulation was decreased in ASK1-depleted hepatocytes (Fig 2C; Appendix Fig S1D), suggesting that ASK1 is involved in the clearance of lipid accumulation in hepatocytes.

Next, we aimed to determine whether ASK1 affects autophagy in cultured hepatocytes. To this end, we assessed levels of LC3-II and p62, two proteins known to be accumulated by blocked autophagy (Bjorkoy *et al*, 2005; Klionsky *et al*, 2012; Gonzalez-Rodriguez *et al*, 2014). Indeed, accumulation of LC3-II and p62 protein levels was increased in ASK1-depleted hepatocytes, indicating blunted autophagy (Fig 2D and Appendix Fig S1E). Consistently, LC3-II punctuates were clearly increased in ASK1-depleted hepatocytes compared to siCtrl cells (Fig 2E). Double immunofluorescence staining revealed increased colocalization of lipid droplets with LC3-II punctuates in ASK1-depleted cells (Fig 2E and Appendix Fig S1F). Of note, the observed heterogeneous individual cell response was confirmed in independent experiments.

To investigate a causative role of autophagy, experiments with the autophagy inhibitor bafilomycin were performed. In agreement with previous experiments, ASK1 knockdown significantly increased LC3 colocalization with lipid droplets in BSA-treated cells (Fig 2F), indicating blunted lipophagy (Singh *et al*, 2009). As expected from previous observation (Pi *et al*, 2015), blockage of autophagy significantly increased LC3 accumulation in siCtrl hepatocytes. However, such effect was clearly blunted in ASK1-depleted cells (siCtrl 227 ± 41% vs. siASK1 49 ± 9%, *P* < 0.01), indicating that impaired autophagy contributes to increased LC3 accumulation in siASK1-treated hepatocytes. Next, lipid accumulation was assessed after bafilomycin treatment. Similar to data presented in Fig 2C, ASK1 depletion significantly increased lipid droplet accumulation in hepatocytes treated with or without palmitate (Fig 2G). Importantly, treatment with bafilomycin significantly enhanced lipid droplet accumulation in siCtrl cells, whereas such effect was blunted in siASK1-treated cells. Taken together, our data indicate that depletion of ASK1 in hepatocytes induces lipid accumulation, potentially via blunting autophagy.

### Hepatocyte-specific ASK1-knockout mice develop NASH and fibrosis

To investigate the role of hepatic ASK1 in liver lipid metabolism *in vivo*, liver-specific ASK1-knockout mice (ASK1^Δhep) were generated using the Cre-lox system (Postic *et al*, 1999). As controls, littermate mice that do not express albumin promoter-driven Cre recombinase (Cre) were used (ASK1^F/F). Western blot analysis confirmed the deletion of ASK1 protein in the liver of ASK1^Δhep mice, whereas its expression was comparable between ASK1^Δhep and ASK1^F/F mice in all other tissues examined (Appendix Fig S2A). To determine the metabolic relevance of hepatocyte-specific ASK1 ablation on NAFLD and liver fibrosis formation, mice were fed either a standard chow or a high-fat diet (HFD) for 20 weeks. While HFD-fed mice revealed higher body weight compared to chow-fed mice, ASK1^Δhep and ASK1^F/F mice showed similar body weight gain, regardless of diet (Appendix Fig 2B). Fat pad mass was similar

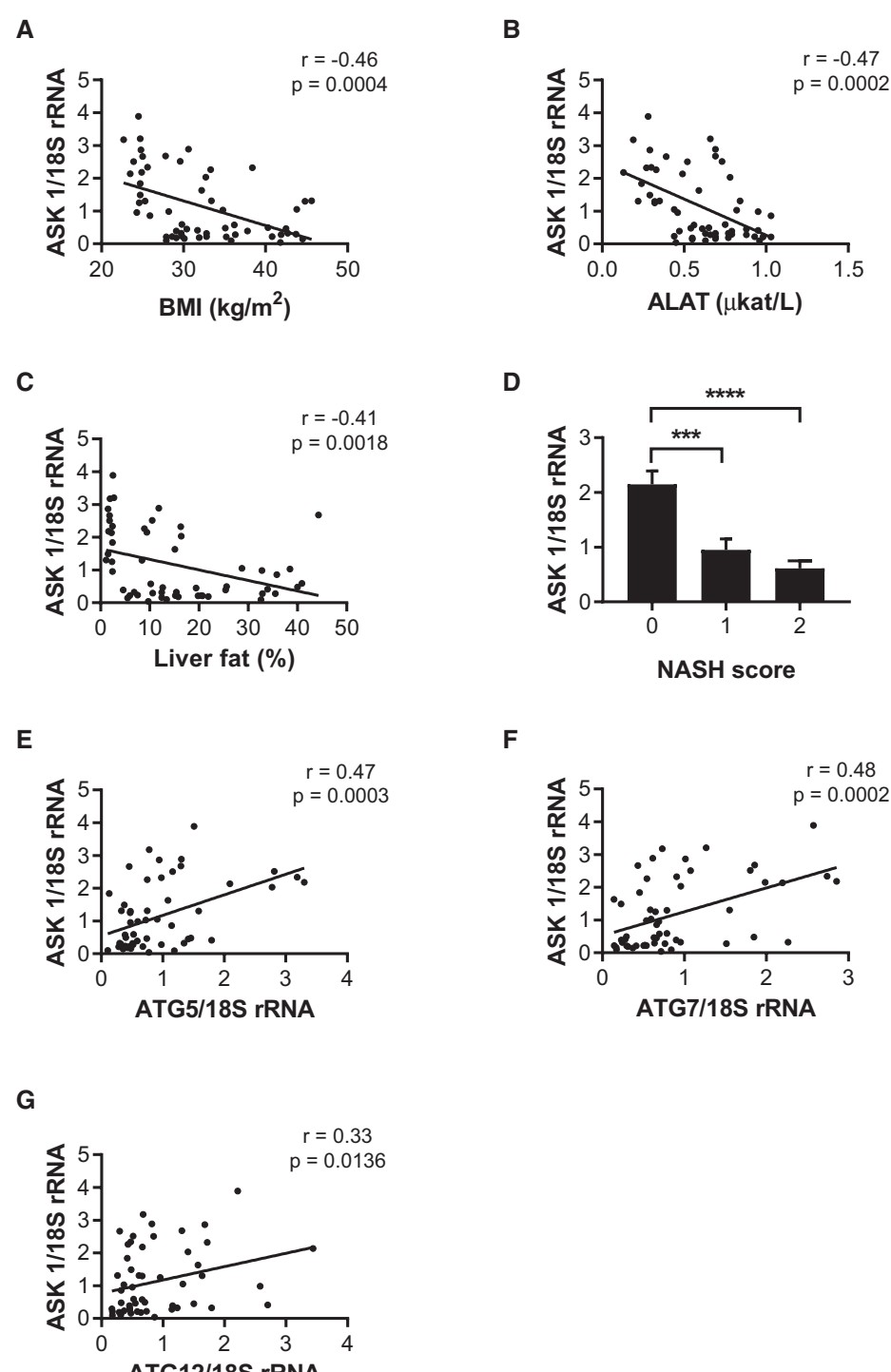

**Figure 1. Decreased liver *ASK1* expression in human subjects with hepatic steatosis and NASH.§**

A–C   Scatter plot and correlation coefficient (*r*) of *ASK1* mRNA expression and BMI (A), plasma alanine aminotransferase (ALAT) (B), and liver fat content (C).

D      *ASK1* mRNA expression in patients with different NASH scores.

E–G   Scatter plot and correlation coefficient (*r*) of *ASK1* mRNA expression and *ATG5* (E), *ATG7* (F), and *ATG12* (G) mRNA expression.

Data information: Data were collected in lean subjects (*n* = 14), obese subjects (*n* = 23), and obese subjects with type 2 diabetes (*n* = 19). Values are expressed as mean ± SEM (D). ***$P < 0.001$, ****$P < 0.0001$. Statistical tests used are as follows: Spearman correlation for (A, B, C, E, F and G); and ANOVA for (D).
Source data are available online for this figure.

---

§Correction added on 9 October 2019, after first online publication: source data for Figure 1 have been added.

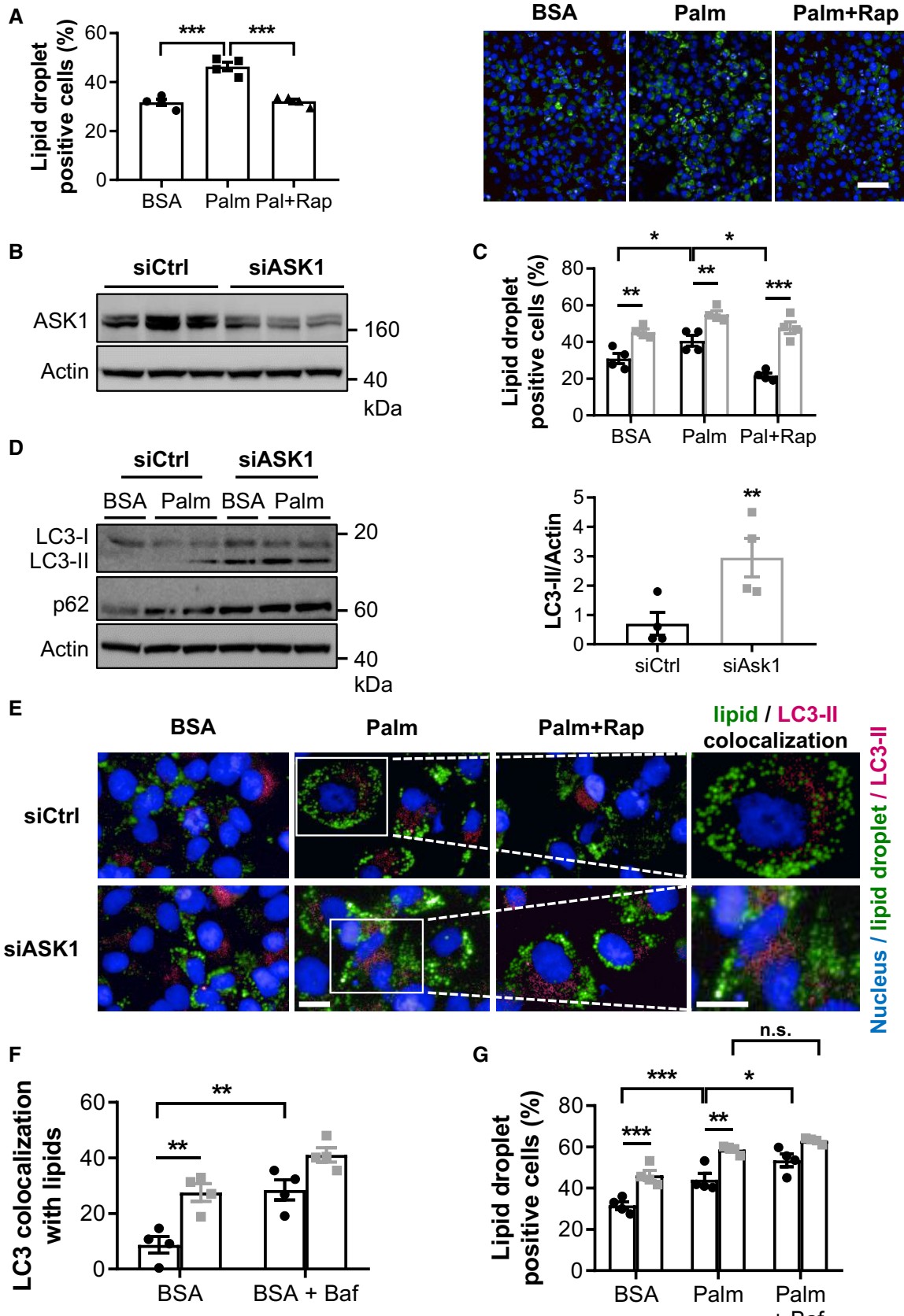

**Figure 2.**

**Figure 2. Depletion of hepatic ASK1 increases lipid storage.**

A  Hepatocytes were treated with BSA, Palm, or Palm + Rap for 24 h. Cells were stained for lipid droplet accumulation (BODIPY 493/503, green) and nuclei (Hoechst, blue). Lipid accumulation was quantified using automated image-based analysis (*n* = 4 biological replicates). Scale bar represents 100 μm.

B  Shown is one representative Western blot from two independent experiments of hepatocytes transfected with siRNA targeting ASK1 (siASK1) or non-targeting siRNA control (siCtrl).

C  Lipid accumulation was quantified using automated image-based analysis in ASK1-knockdown (siASK1; gray bars) or control (siCtrl; black bars) cells treated with BSA, Palm, and Palm + Rap and stained as mentioned above (*n* = 4 biological replicates).

D  Shown is one representative Western blot from two independent experiments (left) and quantification of LC3-II/I and p62 protein levels Palm (right).

E  Colocalization of in siCtrl (*n* = 4 biological replicates) or siASK1 (*n* = 4 biological replicates) cells treated with LC3-II punctate (red) with lipid (BODIPY 493/503, green) and nuclei (Hoechst, blue) in siCtrl or siASK1 cells treated with BSA, Palm, or Palm + Rap. Scale bar represents 100 μm.

F  Colocalization of LC3-II punctate with lipid droplets (BODIPY 493/503, green) was quantified in hepatocytes transfected with siRNA targeting ASK1 (siASK1; gray bars) or non-targeting siRNA control (siCtrl; black bars) and treated with BSA or BSA + Baf for 24 h (*n* = 4 biological replicates).

G  Lipid accumulation was quantified using automated image-based analysis in ASK1-knockdown (siASK1; gray bars) or control (siCtrl; black bars) cells treated with BSA, Palm, or Palm + Baf and stained as mentioned above (*n* = 4 biological replicates).

Data information: Values are expressed as mean ± SEM. *$P < 0.05$; **$P < 0.01$; ***$P < 0.001$. Statistical tests used are as follows: ANOVA for (A, C, F, and G); and *t*-test for (D).

Source data are available online for this figure.

between both genotypes under either diet (Appendix Fig S2C), whereas ASK1$^{\Delta hep}$ mice showed slightly increased liver weight under both chow and HFD compared with ASK1$^{F/F}$ mice (Appendix Fig S2D). Importantly, aggravated hepatic lipid droplet accumulation and significantly higher liver triglyceride (TG) content were observed in HFD-fed ASK1$^{\Delta hep}$ (Fig 3A and B) demonstrating that ASK1 ablation accelerated HFD-induced hepatic steatosis.

In agreement with elevated hepatic steatosis, mRNA expression of hepatic *perilipin* (*Plin*), a gene promoting lipid droplet formation and known to be elevated in fatty livers (Trevino *et al*, 2015), was significantly increased in HFD-fed ASK1$^{\Delta hep}$ compared with ASK1$^{F/F}$ mice (Fig 3C). In contrast, similar expression of genes involved in lipolysis (*Lipe/Hsl*), *de novo* lipogenesis (*Acc1, Fas, and Srebp1*), and β-oxidation (*Cpt1α, Acox,* and *Pparα*) was observed in livers of both genotypes (Appendix Fig S2E). In HFD-fed ASK1$^{\Delta hep}$ mice, expression of hepatic *very low-density lipoprotein receptor* (*Vldlr*) was significantly upregulated (Fig 3C), suggesting that ASK1 inhibition may enhance VLDL uptake or block its secretion, thereby increasing liver triglyceride accumulation.

Hepatic lipid storage is tightly associated with impaired glucose metabolism and chronic hepatic inflammation (Dowman *et al*, 2010; Yang *et al*, 2010). Indeed, HFD-fed ASK1$^{\Delta hep}$ mice showed slightly impaired glucose tolerance (Appendix Fig S2F) as well as increased plasma insulin levels (Appendix Table S2) when compared to ASK1$^{F/F}$ mice indicating that ASK1 deficiency may enhance HFD-induced glucose intolerance. Although 50% higher in knockout animals, circulating insulin levels did not significantly differ between the two genotypes. No difference in glucose tolerance between the two genotypes was observed in age-matched chow-fed mice (Appendix Fig S2F). In addition, expression of pro-inflammatory cytokines and macrophage markers *Il-6, F4/80,* and *Mcp-1(Ccl2)* was elevated in livers of ASK1$^{\Delta hep}$ mice (Fig 3D). In line with increased hepatic inflammation, plasma levels of alanine transaminase (ALT) and aspartate aminotransferase (AST) were markedly elevated in ASK1$^{\Delta hep}$ mice (Appendix Table S2). Hence, ASK1 depletion stimulates hepatic inflammation/injury and consequently the progression of hepatic steatosis to NASH. Circulating cytokine levels were similar between both genotypes (Appendix Table S2), indicating that hepatic ASK1 ablation is mainly influencing local liver inflammation. In addition, plasma glycerol, FFA, TG, and cholesterol levels were comparable (Appendix Table S2). Collectively, these data indicate that hepatic

depletion of ASK1 accelerates HFD-induced development of hepatic steatosis and NASH.

We next wanted to investigate whether ASK1 is involved in the progression of NASH to liver fibrosis (Dowman *et al*, 2010). After 20 weeks of HFD, mRNA expression of *Col1a1* and *Tgfβ1* was significantly higher in livers of HFD-fed ASK1$^{\Delta hep}$ mice (Fig 3E), indicating elevated collagen/fibrogenic cytokine expression in knockout mice. In agreement, collagen deposition as assessed by Sirius Red staining was markedly increased in liver-specific ASK1-knockout mice (Fig 3F and G) revealing the development of liver fibrosis. In addition, immunohistochemical analysis revealed upregulated alpha-smooth muscle actin (α-SMA) protein levels in the liver of HFD-fed ASK1$^{\Delta hep}$ mice (Fig 3H). This suggests that activation of fibrogenic hepatic satellite cells contributed to increased collagen production (Lim *et al*, 2011). In agreement, protein levels of collagen 1A1 were significantly increased in HFD-fed-knockout mice (Fig 3I). To investigate whether liver-expressed ASK1 impacts on age-related development of fatty liver and fibrosis, we analyzed livers of 15-month-old chow-fed control and knockout mice. Indeed, aged ASK1$^{\Delta hep}$ mice revealed elevated hepatic TG levels as well as higher collagen deposition as assessed by Sirius Red staining (Fig 3J and K). In addition, ASK1$^{\Delta hep}$ mice showed impaired glucose tolerance compared to ASK1$^{F/F}$ mice at the age of 15 months (Appendix Fig S3A), indicating that ASK1 deficiency aggravates age-induced glucose intolerance. Hence, liver ASK1 protects from HFD- and age-induced development of NASH and fibrosis.

## Autophagy is impaired in hepatocyte-specific ASK1-knockout mice

In support of an important role of ASK1 in JNK activation, phosphorylation of the latter was significantly reduced in livers of hepatocyte-specific ASK1-depleted mice fed a HFD for 20 weeks (Fig 4A) as was phosphorylation of p38 MAPK (Fig 4A). Activated JNK may activate autophagy via inducing activation of Beclin 1 (BCN1; McEwan & Dikic, 2011; Gade *et al*, 2014). In line with reduced p-JNK protein levels, phosphorylation of BCN1 was downregulated in ASK1$^{\Delta hep}$ compared to ASK1$^{F/F}$ mice (Fig 4A). Moreover, accumulation of LC3-II and aggregation of autophagy substrate p62 protein levels were markedly elevated in ASK1$^{\Delta hep}$ mice (Fig 4A) indicating impaired hepatic autophagy. In agreement, protein levels of the

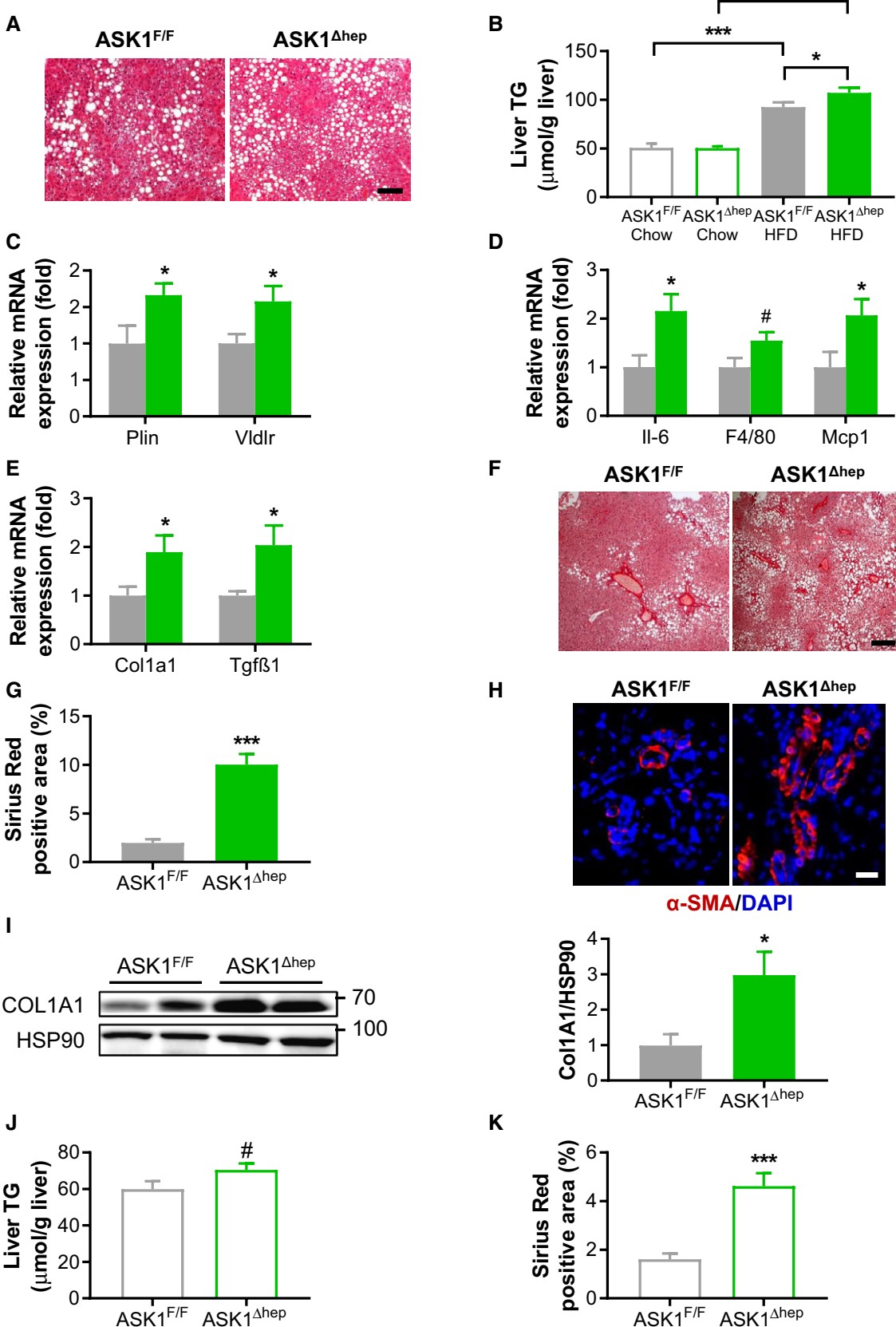

Figure 3.

**Figure 3. Hepatocyte-specific ASK1-knockout mice develop NASH and fibrosis.**

A   Representative images of liver sections stained with H&E from mice fed a HFD for 20 weeks. Scale bar represents 100 μm.

B   Liver triglyceride (TG) content of mice fed a chow (ASK1$^{F/F}$ $n$ = 6 mice; ASK1$^{\Delta hep}$ $n$ = 6 mice) or HFD (ASK1$^{F/F}$ $n$ = 14 mice; ASK1$^{\Delta hep}$ $n$ = 15 mice).

C   Relative mRNA expression of *perilipin* (*Plin*) and *very low-density lipoprotein receptor* (*Vldlr*) in liver of mice fed a HFD for 20 weeks [ASK1$^{F/F}$ $n$ = 6 mice; ASK1$^{\Delta hep}$ $n$ = 7 mice (*Vldlr*) or $n$ = 8 mice (*Plin*)].

D   Relative mRNA expression of genes involved in inflammation in liver of mice fed a HFD for 20 weeks (ASK1$^{F/F}$ $n$ = 6 mice; ASK1$^{\Delta hep}$ $n$ = 7 mice (F4/80, Mcp1) or ASK1$^{\Delta hep}$ $n$ = 8 mice (Il-6)).

E   Relative mRNA expression of *collagen, type I, alpha 1* (*Col1a1*) (ASK1$^{F/F}$ $n$ = 11 mice; ASK1$^{\Delta hep}$ $n$ = 15 mice), and *transforming growth factor beta 1* (*Tgfβ1*) (ASK1$^{F/F}$ $n$ = 6 mice; ASK1$^{\Delta hep}$ $n$ = 8 mice) in liver of HFD-fed mice.

F, G   (F) Representative images of liver sections stained with Sirius Red (scale bar represents 100 μm) and (G) quantification of Sirius Red-positive area in liver of mice fed a HFD (ASK1$^{F/F}$ $n$ = 8 mice; ASK1$^{\Delta hep}$ $n$ = 8 mice).

H   Representative images of liver sections stained with α-SMA from mice fed a HFD for 20 weeks. Scale bar represents 100 μm.

I   Shown is one representative Western blot from two independent experiments (left) and quantification of hepatic COL1A1 protein levels in mice fed a HFD for 20 weeks (ASK1$^{F/F}$ $n$ = 8 mice; ASK1$^{\Delta hep}$ $n$ = 8 mice) (right).

J, K   Liver triglyceride (TG) content (ASK1$^{F/F}$ $n$ = 7 mice; ASK1$^{\Delta hep}$ $n$ = 7 mice) and quantification of Sirius Red-positive area in livers (ASK1$^{F/F}$ $n$ = 7 mice; ASK1$^{\Delta hep}$ $n$ = 7 mice) of 15-month-old chow-fed mice.

Data information: Values are expressed as mean ± SEM. $^{\#}P$ = 0.08; $^{*}P$ < 0.05; $^{***}P$ < 0.001. Statistical tests used as follows: ANOVA for (B); and *t*-test for (C, D, E, G, I, J, and K). Source data are available online for this figure.

autophagy marker ATG12 were significantly reduced further indicating blunted autophagy in knockout mice (Fig 4A). The latter was further verified by LC3-II immunostaining. Consistent with increased LC3-II protein levels, LC3-II punctuates were higher in liver sections of ASK1$^{\Delta hep}$ compared with ASK1$^{F/F}$ mice (Fig 4B), suggesting that HFD feeding impairs autophagy in liver-specific ASK1-deficient mice. Similarly, accumulation of LC3-II and p62 protein levels was significantly elevated in 15-month-old ASK1$^{\Delta hep}$ mice (Appendix Fig S3B–D). Collectively, these data indicate that hepatic ASK1 depletion accelerates HFD-induced blockage of autophagy, potentially contributing to impaired clearance of liver lipids.

**Depletion of ASK1 impairs autophagy in primary hepatocytes**

To determine whether reduced autophagy in ASK1-depleted hepatocytes contributes to elevated lipid accumulation, experiments in primary hepatocytes treated with the autophagy inhibitor bafilomycin were performed. In line with experiments in HepG2 cells, rapamycin significantly decreased palmitate-induced lipid droplet accumulation in hepatocytes isolated from ASK1$^{F/F}$ but not from ASK1$^{\Delta hep}$ mice (Fig 5A). Bafilomycin blunted the effect of rapamycin on the clearance of palmitate-induced lipid droplet accumulation only in hepatocytes isolated from control mice (Fig 5A), indicating that blunted autophagy in ASK1-depleted hepatocytes contributes to elevated lipid accumulation. Of note, LC3-II punctuates were increased in ASK1-depleted hepatocytes treated with BSA compared to control cells (Fig 5B), further suggesting that lack of ASK1 inhibits autophagy. In line with experiments performed in HepG2 cells (Fig 2E), treatment with rapamycin reduced palmitate-induced lipid droplet colocalization with LC3-II punctuates in hepatocytes isolated from control mice, whereas it had no effect in ASK1-depleted cells. As expected, the autophagy inhibitor bafilomycin increased LC3-II punctuates in control cells (Fig 5B).

**Pharmacological inhibition of ASK1 aggravates HFD-induced hepatic lipid accumulation**

Given the suggested role of hepatic ASK1 ablation in the pathogenesis of NAFLD, we evaluated the therapeutic effects of pharmacological ASK1 inhibition. C57BL/6J mice fed a HFD for 11 weeks were treated with or without an ASK1 inhibitor for the last 5 weeks of HFD feeding. Mice treated with the ASK1 inhibitor showed similar body weight gain compared to control mice (data not shown). As expected, treatment with the ASK1 inhibitor significantly decreased hepatic ASK1 phosphorylation compared to control mice (Fig 6A). In line with results obtained from ASK1$^{\Delta hep}$ mice, treatment with the ASK1 inhibitor markedly aggravated HFD-induced hepatic lipid droplet accumulation and liver TG content (Fig 6B and C). In parallel, higher fasting blood glucose and increased plasma FFA and TG levels were detected in mice treated with the ASK1 inhibitor (Appendix Table S3). Pharmacological inhibition of ASK1 markedly downregulated activation of the JNK-autophagy axis, as evidenced by reduced phosphorylation of JNK and BCN1, accompanied by increased LC3-II and p62 protein levels compared to control mice (Fig 6A). Taken together, pharmacological inhibition of ASK1 impairs hepatic autophagy and increases HFD-induced liver lipid accumulation.

**Liver-specific ASK1 overexpression ameliorates HFD-induced steatosis and protects from CCl4-induced fibrosis**

To investigate a regulatory role of liver ASK1 in the development of steatosis and fibrosis, mice with liver-specific overexpression of ASK1 were generated (ASK1$^{+hep}$). After Cre-lox-mediated excision of the stop cassette, such mice express ASK1 under the ROSA26 promoter (Casola, 2010). Accordingly, albumin promoter-driven Cre expression promotes overexpression of ASK1 specifically in the liver (Appendix Fig S4A). ASK1$^{+hep}$ and control littermate mice (ASK1$^{F/F}$) were fed a HFD for 20 weeks. Glucose tolerance was significantly improved in HFD-fed liver-specific ASK1 overexpressing compared to control mice (AUC ASK1$^{F/F}$ 2,267 ± 195 mmol/l*min vs. ASK1$^{+hep}$ 1,615 ± 116 mmol/l*min; $P$ < 0.05). Lower hepatic lipid droplet accumulation and significantly reduced liver TG content were observed in HFD-fed ASK1$^{+hep}$ compared with control mice (Fig 7A and B), suggesting that ASK1 overexpression blunts HFD-induced hepatic steatosis. In parallel, autophagy was elevated in HFD-fed ASK1$^{+hep}$ mice (Fig 7C and Appendix Fig S4B). Next, we wanted to elaborate whether ASK1 overexpression protects from carbon tetrachloride (CCl4)-induced fibrosis (Mederacke *et al*, 2013). Indeed, collagen deposition as assessed by Sirius

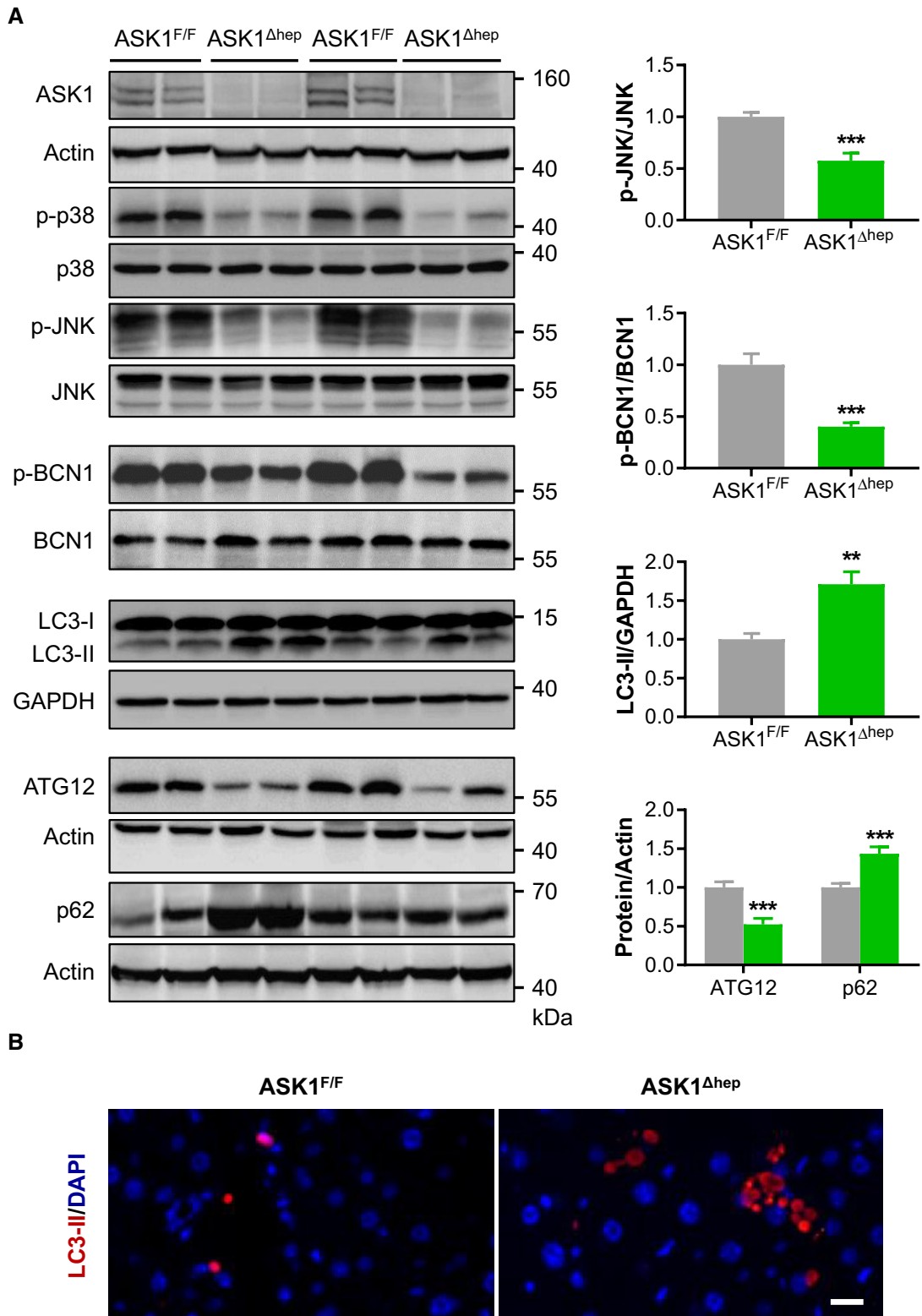

**Figure 4. Autophagy is impaired in hepatocyte-specific ASK1-knockout mice.**

A   Shown is one representative Western blots from two independent experiments (left) and quantification of respective protein levels in mice fed a HFD for 20 weeks (ASK1$^{F/F}$ $n$ = 8 mice; ASK1$^{\Delta hep}$ $n$ = 8 mice) (right). Values are expressed as mean ± SEM. **$P$ < 0.01; ***$P$ < 0.001. Statistical test used is as follows: $t$-test.

B   Representative immunofluorescence images of liver sections stained for LC3-II punctate (red) and nuclei (DAPI, blue) in mice. Scale bar represents 100 μm.

Source data are available online for this figure.

## A

### Primary hepatocytes

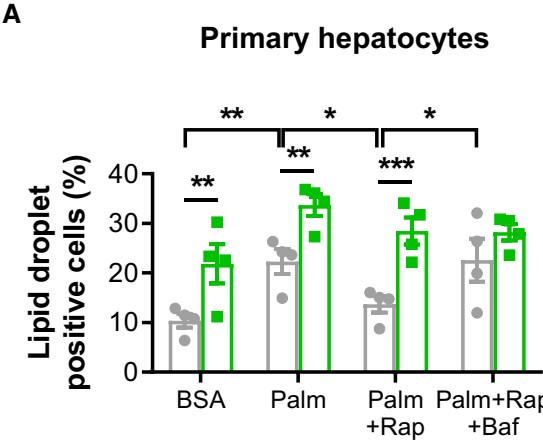

## B

### Primary hepatocytes

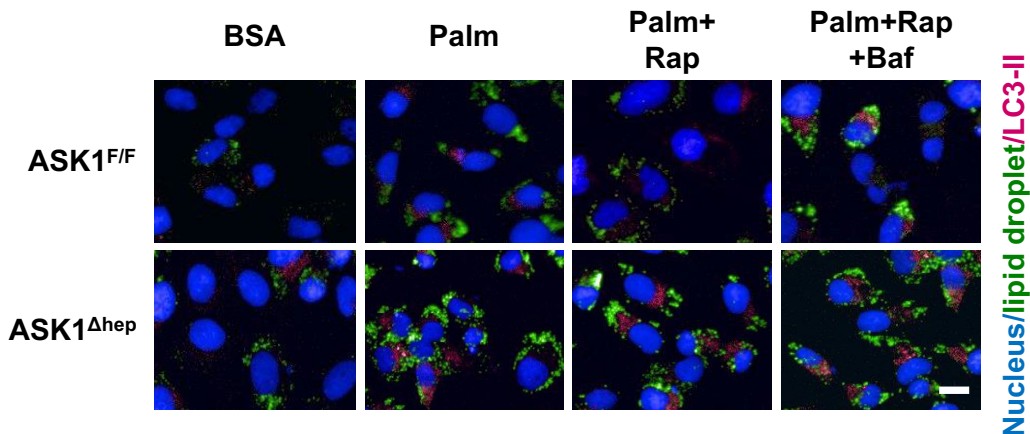

**Figure 5. Depletion of ASK1 impairs autophagy in primary hepatocytes.**

A Primary hepatocytes isolated from chow-fed ASK1[F/F] (grey bars) and ASK1[Δhep] (green bars) mice were treated with BSA, Palm, Palm + Rap, or Palm + Rap + Baf for 24 h. Cells were stained for lipid droplet accumulation (BODIPY 493/503, green) and nuclei (Hoechst, blue). Lipid accumulation was quantified using automated image-based analysis ($n$ = 4 biological replicates). Values are expressed as mean $\pm$ SEM. *$P < 0.05$; **$P < 0.01$; ***$P < 0.001$. Statistical test used is as follows: ANOVA.

B Colocalization of LC3-II punctate (red) with lipid (BODIPY 493/503, green) and nuclei (Hoechst, blue) in primary hepatocytes treated with BSA, Palm, Palm + Rap, or Palm + Rap + Baf. Scale bar represents 100 μm.

Red staining was significantly lower in livers of CCl4-injected ASK1[+hep] mice (Fig 7D). In parallel, mRNA expression of pro-inflammatory cytokines as well as macrophage and fibrosis markers including *Col1a1* and *Tgfβ1* was reduced (Fig 7E and F). Taken together, liver-specific overexpression of ASK1 protects from the development of HFD-induced steatosis and CCl4-induced fibrosis.

## Discussion

The present study uncovers a protective role of liver-expressed ASK1 in the development of NAFLD and liver fibrosis. Such notion is supported by the fact that genetic depletion and pharmacological inhibition of ASK1 *in vivo* and *in vitro* increased hepatic lipid droplet accumulation and/or liver fibrosis, potentially via blocking autophagy function. In line, liver-specific ASK1 overexpression protected

mice from HFD-induced steatosis and CCl4-induced fibrosis. In human livers of lean and obese subjects, *ASK1* expression was negatively associated with liver fat content and NASH scores, but correlated positively with the autophagy markers ATG5, ATG7, and ATG12.

Autophagy plays an important role in hepatic lipid homeostasis. In fact, lipid droplets are substrates of autophagy, since they are sequestered in autophagosomes for degradation within lysosomes (Singh *et al*, 2009). In HFD-fed ASK1[Δhep] mice, we found decreased autophagy as determined by reduced protein levels of ATG12 as well as increased accumulation of LC3-II and p62. Of note, activation of autophagy decreases accumulation of LC3-II and p62, whereas blockage of autophagy increases LC3-II and p62 protein levels (Bjorkoy *et al*, 2005; Klionsky *et al*, 2012; Gonzalez-Rodriguez *et al*, 2014). Reduced autophagy in ASK1[Δhep] mice was paralleled by elevated liver lipid accumulation indicating that

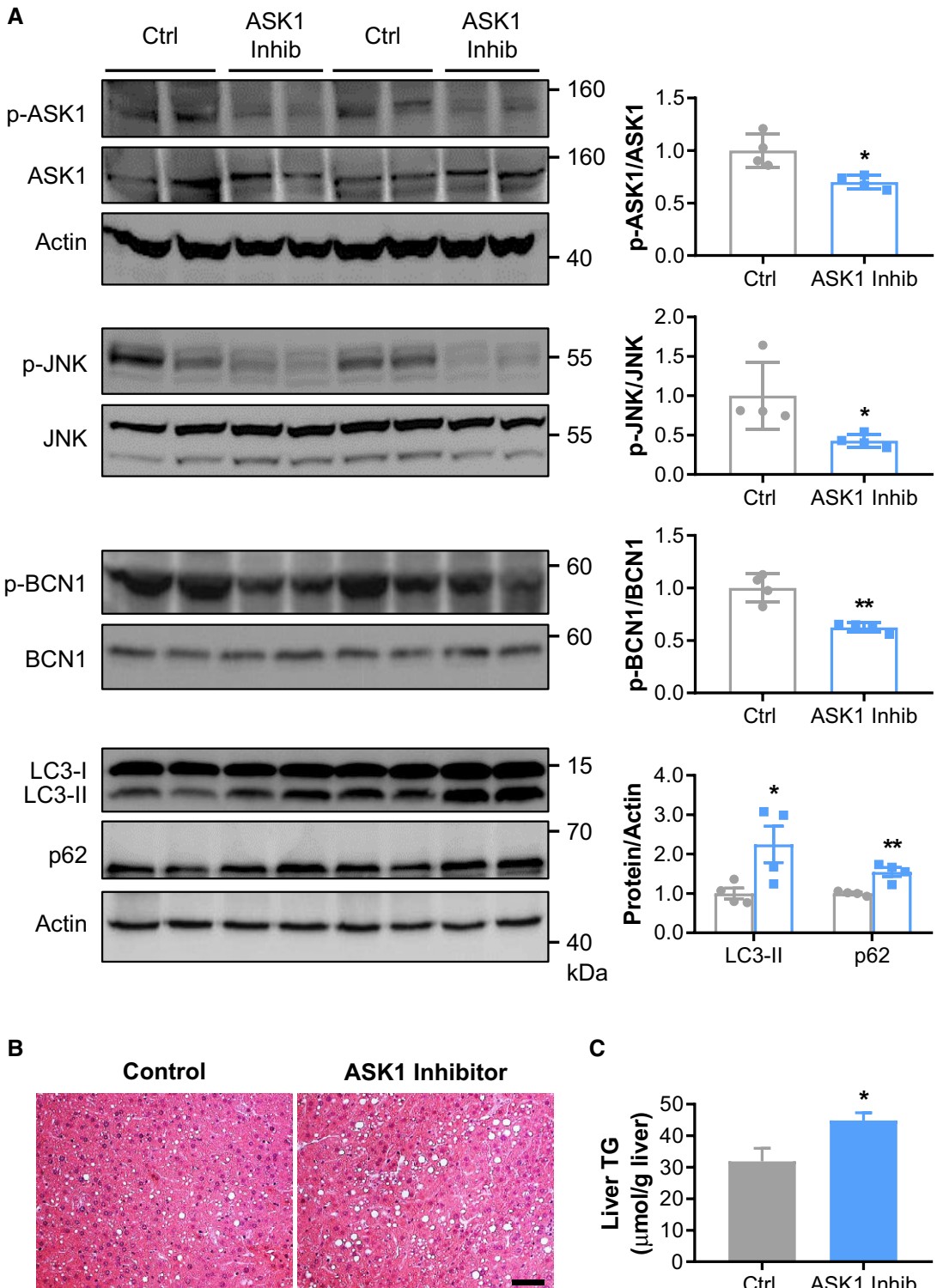

**Figure 6. Pharmacological inhibition of ASK1 induces hepatic lipid accumulation.**

A    Western blots (left) and quantification of respective protein levels in liver of HFD-fed mice treated with or without ASK1 inhibitor ($n = 4$ mice per treatment) (right).

B, C   (B) Representative images of liver sections stained with H&E (scale bar represents 100 μm) and (C) liver triglyceride (TG) content of HFD-fed mice treated with or without ASK1 inhibitor ($n = 8$ mice per treatment).

Data information: Values are expressed as mean ± SEM. *$P < 0.05$; **$P < 0.01$. Statistical test used is as follows: $t$-test for (A, C).
Source data are available online for this figure.

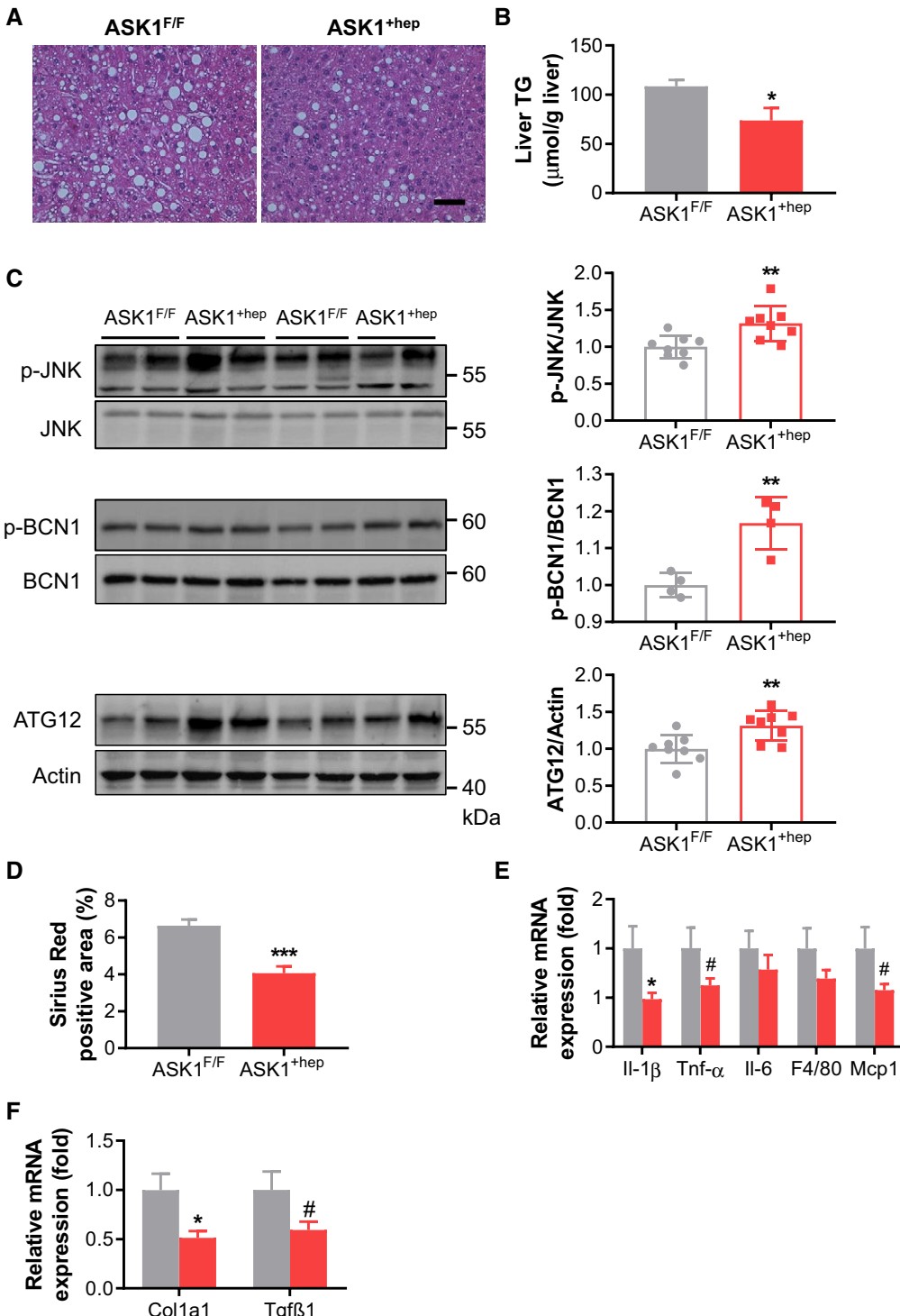

**Figure 7. Liver-specific ASK1 overexpression ameliorates HFD-induced steatosis and protects from CCl4-induced fibrosis.**

A Representative images of liver sections stained with H&E from mice fed a HFD for 20 weeks. Scale bar represents 100 μm.

B Liver triglyceride (TG) content (ASK1$^{F/F}$ $n$ = 11 mice; ASK1$^{+hep}$ $n$ = 10 mice) in livers of mice fed a HFD for 20 weeks.

C Western blots (left) and quantification of protein levels of respective targets in liver of HFD-fed mice ($n$ = 8 mice per genotype for p-JNK and ATG12, $n$ = 4 mice per genotype for pBCN1) (right).

D Quantification of Sirius Red-positive area (ASK1$^{F/F}$ $n$ = 11 mice; ASK1$^{+hep}$ $n$ = 9 mice) in livers of CCl4-injected mice.

E, F Relative mRNA expression (ASK1$^{F/F}$ $n$ = 13 mice; ASK1$^{+hep}$ $n$ = 14 mice) of respective genes involved in inflammation and fibrosis in liver of CCl4-injected mice.

Data information: Values are expressed as mean ± SEM. $^{\#}P < 0.1$; $^{*}P < 0.05$; $^{**}P < 0.01$; $^{***}P < 0.001$. Statistical test used is as follows: $t$-test for (B–F).
Source data are available online for this figure.

blunted autophagy may be critically involved in the observed phenotype. In support for an important role of autophagy in ASK1's effect on hepatic lipid accumulation, blockage of autophagy affected lipid accumulation in control cells but not in hepatocytes with siRNA or Cre-lox-mediated ASK1 depletion. Moreover, colocalization of lipid droplets with LC3 punctuates was increased in ASK1-depleted cells, indicating blunted lipophagy. As previously reported (Wang et al, 2017a), lipid droplets and LC3 puncta in hepatocytes were present in parallel rather than overlapping. In line with findings in mice with genetic ASK1 depletion, pharmacological inhibition of ASK1 accelerated HFD-induced lipid droplet storage and liver TG content in C57BL/6J mice. In the latter mice, elevated steatosis was accompanied by enhanced accumulation of LC3-II and p62 protein levels suggesting impaired hepatic autophagy. Similarly, ASK1 knockdown in hepatocytes in vitro resulted in higher lipid accumulation and increased accumulation of LC3-II and p62 protein levels.

The ASK1 downstream target JNK was shown to regulate autophagy via activation of BCN1 (McEwan & Dikic, 2011). In HFD-fed mice, we found reduced p-JNK and p-BCN1 levels after genetic as well as pharmacological inhibition of ASK1, indicating that ASK1 regulates hepatic lipid accumulation via the JNK-autophagy axis. In support of latter axis, JNK inhibition blunted autophagy in hepatocytes (Zhang et al, 2016a; Wang et al, 2018). In agreement with our findings, hepatic JNK1-knockout mice develop hepatic steatosis (Sabio et al, 2009). In such mice, impaired VLDL assembly contributed to elevated lipid accumulation. Moreover, a previous study has reported increased hepatic lipid accumulation induced by VLDLR overexpression, while VLDLR-deficient mice were protected from hepatic steatosis induced by HFD feeding (Jo et al, 2013). Furthermore, activation of autophagy degrades VLDL assembly to decrease hepatic lipid storage (Conlon et al, 2016; Zamani et al, 2016). Hence, increased expression of Vldlr in HFD-fed ASK1$^{\Delta hep}$ mice may contribute to elevated hepatic steatosis in ASK1-knockout mice via reduced activation of the JNK-autophagy axis.

Besides affecting lipid accumulation (Singh et al, 2009; Gonzalez-Rodriguez et al, 2014), defective autophagy may also be involved in the development of NASH and liver fibrosis (Inokuchi-Shimizu et al, 2014; Lodder et al, 2015). In fact, mice lacking ATG5 in macrophages develop elevated liver injury and fibrosis induced by carbon tetrachloride treatment due to increased Il-1β expression (Lodder et al, 2015). Such data suggest that autophagy suppresses liver fibrosis by regulating inflammatory pathways. Moreover, liver-specific ATG7 and ATG5-deficient mice developed multiple liver tumors due to p62 accumulation (Takamura et al, 2011). In line with an important role of autophagy in the development of fibrosis and NASH, we report herein that hepatocyte-specific ASK1-knockout mice develop liver fibrosis as evidenced by high collagen deposition, elevated protein levels of α-SMA, and increased expression of Col1a1. Fibrosis formation was paralleled by elevated expression of inflammatory cytokines such as Il-6 and Tgfβ1. Thus, ASK1 suppresses the formation of NASH and fibrosis formation, potentially via activation of autophagy.

Elevated liver lipid accumulation in ASK1$^{\Delta hep}$ mice was observed after 20 weeks of HFD feeding. Of note, liver steatosis was not enhanced in ASK1$^{\Delta hep}$ mice after a shorter period of HFD, i.e., 6 weeks (Appendix Fig S5A). In these mice, liver inflammation (i.e.,

lower Tnf-α expression) was reduced and glucose tolerance improved when compared to control littermates (Appendix Fig S5B and C). Consequently, ASK1 depletion may have a beneficial effect in an earlier phase of high-fat feeding, whereas it negatively affects liver lipid metabolism and promotes NASH after prolonged high-fat diet intake.

In contrast to our findings, a recent study reported a positive effect of hepatic ASK1 (and JNK1) ablation in the development of steatohepatitis in HFD-fed mice (Wang et al, 2017b; Zhang et al, 2018). Such discrepancy may be partially explained by differences in the experimental setup as we used a HFD mainly consisting of coconut oil, while Wang et al (2017b) used a HFD containing lard. In addition, sucrose content was 50% higher in the diet used in the present study. Based on the phenotypic transformation observed in our mouse model depending on HFD exposure time, it would be interesting to investigate whether a longer period of HFD feeding induces deleterious effects in aforementioned mouse models (Zhang et al, 2016a; Wang et al, 2017b). Recent clinical trials have identified decreased autophagy markers such as elevated numbers of LC3-II puncta and increased protein levels of p62 in liver of patients with steatosis or NASH (Gonzalez-Rodriguez et al, 2014). Such data suggest that autophagy may also be critically involved in the development of hepatic steatosis and NASH in humans. Indeed, molecules targeting autophagy revealed beneficial effects on NASH both in animal and clinical studies (Allaire et al, 2019). However, tested drugs may have affected other pathways than autophagy and therefore suffered from specificity toward the latter. Therefore, identification of more selective and hepatocyte-specific activators of autophagy to tackle NASH is warranted. In line with a role of autophagy in NASH, we found significant negative correlations between liver ASK1 expression and liver fat content as well as circulating ALAT levels in human subjects. Moreover, decreased ASK1 expression was associated with increasing NASH scores, whereas ASK1 correlated positively with the autophagy markers ATG5, ATG7, and ATG12. These data support a protective role for liver-expressed ASK1 in the development of hepatic steatosis as well as NASH in humans. In contrast, treatment with the ASK1 inhibitor selonsertib (in combination with the humanized monoclonal antibody directed against lysyl oxidase-like molecule 2 simtuzumab) was recently reported to reduce liver fibrosis in a subgroup of patients with manifest NASH and moderate-to-severe fibrosis (Loomba et al, 2018), suggesting that ASK1 inhibition may be a useful strategy to halt or partly reverse evolved liver pathologies at least in some patients. Although both selonsertib and the ASK1 inhibitors used in this study (MSC 2032964A) are potent and selective ASK1 inhibitors, their different molecular structure may explain observed differences. In addition, ASK1 may not only be inhibited in the liver but also be inhibited in other organs such as adipose tissue. Of note, we have evidence that inhibition of ASK1 in white adipose tissue may promote browning and, thus, may reduce hepatic lipid accumulation and improve glucose metabolism. Clearly, further studies investigating the role of ASK1 in the development and/or reversal of NASH and fibrosis are warranted.

Taken together, we identify a beneficial role for liver-specific ASK1 expression in the prevention of obesity-associated hepatic steatosis and liver fibrosis potentially through induction of autophagy that may have translational therapeutic implications to tackle the development of NAFLD and liver fibrosis in humans.

# Materials and Methods

### Human studies

The expression of ASK1 and ATG5, ATG7 and ATG12 mRNAs was measured in livers of lean subjects and patients who underwent open abdominal surgery for Roux-en-Y bypass, sleeve gastrectomy, or elective cholecystectomy. Liver biopsies were taken during surgery, immediately snap-frozen in liquid nitrogen, and stored at −80°C until further preparation. Measurement of total body and liver fat, tissue sample handling, and analysis of blood samples including measurement of serum adiponectin and leptin concentrations has been performed as described previously (Kloting *et al*, 2010; de Guia *et al*, 2015). Serum alanine aminotransferase (ALAT) was photometrically measured using the Cobas C 6000 System (Roche, Germany). All investigations have been approved by the ethics committee of the University of Leipzig (363-10-13122010 and 017-12-230112) and were carried out in accordance with the Declaration of Helsinki, and all study participants provided witnessed written informed consent before entering the study. Expression of human ASK1, ATG5, ATG7, and ATG12 mRNAs was measured by qRT–PCR using Assay-on-Demand gene expression kits (*ASK1* (Hs01039896_m1), *ATG5* (Hs00169468_m1), *ATG7* (Hs00197348_m1), and *ATG12* (Hs04980076_s1); Applied Biosystems, Darmstadt, Germany), and fluorescence was detected on an ABI PRISM 7000 Sequence Detector (Applied Biosystems). Expression of ASK1, ATG5, ATG7, and ATG12 mRNAs was calculated relative to the expression of 18S rRNA (Hs99999901_s1; Applied Biosystems).

### Materials and reagents

Sodium palmitate (Palm), bovine serum albumin (BSA), rapamycin (Rap), bafilomycin A1 (Baf), and carbon tetrachloride (CCl4) were obtained from Sigma-Aldrich (Buchs, Switzerland). BODIPY 493/503, ASK1 inhibitor (MSC 2032964A), was purchased from Bio-Techne AG (Zug, Switzerland); Hoechst and Lipofectamine® 2000 were purchased from Invitrogen (Reinach, Switzerland). RNA extraction kits were obtained from Promega (Dübendorf, Switzerland). QPCR master mix was purchased from Promega. SYBR green was obtained from Roche Diagnostics (Rotkreuz, Switzerland).

### Animals

To obtain mice with a conditional knockout allele of ASK1, exon 14 was targeted. Embryonic stem (ES) cell clone with exon 14 flanked by loxP site and a FRT-flanked selection cassette were obtained from the European Conditional Mouse Mutagenesis Program (EUCOMM). Morula aggregation for producing chimeric mice from ES cells (B6 background, TyrC+) and foster mothers (Tgv, TyrC−) was performed at Institute of Laboratory Animal Science, University of Zurich. Then, chimeric mice were crossed with B6-albino mice (TyrC−). Their offspring with successful germ-line transmission (black mice) were screened for the presence of the inserted loxP sites by PCR. Mice heterozygous for floxed ASK1 were subsequently inbred to produce homozygous mutant mice. The FRT-flanked selection cassette was removed by crossbreeding the floxed ASK1 mice with Flpo deleter mice (Kranz *et al*, 2010).

Hepatocyte-specific ASK1-knockout (ASK1$^{flox/flox}$, Alb-Cre$^{+/−}$; ASK1$^{\Delta hep}$) mice were generated by crossing homozygous ASK1 floxed (ASK1$^{flox/flox}$, ASK1$^{F/F}$) mice with mice carrying the Cre recombinase transgene under the control of the albumin promoter (C57BL/6J-Tg (Alb-Cre) mice). Littermate mice (ASK1$^{flox/flox}$, Alb-Cre$^{−/−}$; ASK1$^{F/F}$) without albumin-Cre recombinase (Cre) expression were used as a control groups. Liver-specific ASK1-overexpressing mice were generated in collaboration with the company PolyGene (Rümlang, Switzerland). The cDNA of ASK1 was inserted into the ROSA26 locus together with a loxP-flanked stop cassette (ROSA26-ASK1$^{F/F}$). ASK1 overexpression in the liver was induced via Cre-lox-mediated excision of the stop cassette by breeding ROSA26-ASK1$^{F/F}$ mice to aforementioned Alb-Cre mice (ROSA26-ASK1$^{+hep}$).

All animal studies were conformed to the Swiss animal protection laws and were approved by the Cantonal Veterinary Office in Zurich, Switzerland. Six- to 26 week-old-male mice were used for experiments. Mice were housed in a pathogen-free animal facility on a 12-h dark/light cycle and fed *ad libitum* with regular chow diet (Provimi Kliba, Kaiseraugst, Switzerland) or a high-fat diet (58 kcal% fat w/sucrose Surwit diet, D12331, Research Diets, New Brunswick, NJ, USA). The latter was started at the age of 6 weeks and maintained for either 6 or 20 weeks. In experiments, chow-fed control mice were at the same age as HFD-fed mice. Mice receiving the ASK1 inhibitor were allocated to groups based on body weight (similar mean ± SD in starting body weight between the groups).

### Genotyping of animals

All mice were genotyped by PCR with primers amplifying the Cre transgene (generating 310 bp Cre allele products and 510 bp control) and ASK1 (generating 191 bp wild-type and 216 bp "floxed" allele products).

### ASK1 inhibitor administration

ASK1 inhibitor was administered as previously described (Guo *et al*, 2010). In brief, ASK1 inhibitor was dissolved in 0.5% carboxymethylcellulose/0.25% Tween 20 in distilled water. Male C57BL/6J littermate mice fed a HFD for 11 weeks were treated either with ASK1 inhibitor (30 mg/kg body weight) or with control (0.5% carboxymethylcellulose/0.25% Tween 20 dissolved in distilled water) during the last 5 weeks of HFD feeding by oral gavage once/day for 5 days per week. ASK1 inhibitor-treated mice were fasted for 5 h before tissue sampling.

### CCl4 administration

Carbon tetrachloride (CCl4; 0.5 µl/g body weight, dissolved in corn oil at a ratio of 1:3) was intraperitoneally (i.p.) injected three times per week for 6 weeks.

### Liver triglyceride measurement

Mice were fasted overnight for tissue sampling. Hepatic triglyceride (TG) was extracted from 50 to 100 mg liver by using chloroform and methanol, 2:1 (v/v; Wuest *et al*, 2016), and TG concentration was measured with an enzymatic assay kit (Roche Diagnostics).

ASK1 inhibitor-treated mice were fasted for 5 h before tissue sampling.

## Glucose tolerance tests

Mice were fasted overnight, and baseline glucose levels were measured. Thereafter, glucose (2 g/kg body weight) was injected intraperitoneally (i.p.) and blood glucose concentration was measured from tail-tip blood after 15, 30, 45, 60, 90, and 120 min by using a glucometer (Accu-Chek Aviva; Roche Diagnostics).

## Metabolic plasma analysis

Individual mouse body weights were monitored every week. Mice were fasted overnight for blood sampling. ASK1 inhibitor-treated mice were fasted for 5 h before blood sampling. Free fatty acid (FFA), glycerol, triglyceride (TG), total cholesterol (TC) alanine transaminase (ALT), and aspartate aminotransferase (AST) levels were measured by Roche Cobas Mira (Sawgrass Drive, Bellport, NY, USA). The commercial kits were obtained from Diatools AG (Villmergen, Switzerland). Insulin was determined as previously described (Konrad et al, 2007). Plasma cytokine tumor necrosis factor alpha (TNF-α), interleukin 10 (IL-10), interleukin 6 (IL-6), and keratinocyte chemoattractant (KC/CXCL1) were measured using MSD technology (Meso Scale Discovery, Gaithersburg, MD, USA). All analyses were carried out according to manufacturer's protocols.

## Histological analysis

Liver tissue was fixed in 4% formaldehyde solutions and embedded in paraffin. Liver sections (4 μm thick) were stained with hematoxylin and eosin (H&E) or Sirius red (Sakaida et al, 2004). Sirius red-positive areas were measured and quantified using ImageJ software (Inokuchi-Shimizu et al, 2014).

## Immunofluorescence

Liver sections (4 μm thick) were blocked and incubated overnight with primary LC3A/B (#12741, Cell Signaling Technology, Leiden, the Netherlands) or anti-actin α-smooth muscle (A5228, Sigma-Aldrich) antibody and a corresponding secondary Cy3 antibody. Immunofluorescence images were obtained by using a fluorescence microscope (Axioplan 2 imagining) with AxioVision software (version 4.6).

## Cell culture

Human hepatocellular carcinoma HepG2 cells were plated on black 96-well plates with clear bottoms at $1.2 \times 10^4$ cells/well in DMEM containing 1 g/l D-glucose, L-glutamine, pyruvate, 10% fetal bovine serum (FBS), and 1% antibiotics. After 48 h, 60–80% confluent cells were treated with or without 0.4 mM BSA, 0.4 mM palmitate, and 1 mM rapamycin for 24 h in serum- and antibiotic-free media. The cells were fixed in 4% formaldehyde solution and stained for lipid droplet accumulation (BODIPY 493/503) and nuclei (Hoechst). Fluorescence images were taken with the Operetta high-throughput imaging system, and lipid droplet accumulation was quantified

using automated image-based analysis with Harmony software (Meissburger et al, 2011; Challa et al, 2012, 2015).

## Cell transfection

For the RNA interference assay, 15 μl Opti-Mem and 200 nM small interfering RNA targeting ASK1, siRNA (siASK1), or control siRNA (siCtrl) were mixed and 60–70% confluent cultured HepG2 cells were transfected using Lipofectamine® 2000 with antibiotic-free media. After 24 h, the medium was changed and the cells were treated with or without 0.4 mM BSA, 0.4 mM palmitate, and 1 mM rapamycin in serum-free medium. Then after 24 h of treatment, the cells were fixed with 4% formaldehyde, blocked, and incubated O/n with primary LC3A/B antibody (#12741, Cell Signaling Technology) and Cy3 anti-rabbit secondary antibody was used for LC3-II punctuate staining. Lipid droplets were stained as mentioned above. Fluorescent images were taken, and automated lipid droplet quantification was performed with the Operetta high-throughput imaging system (Meissburger et al, 2011; Challa et al, 2012, 2015). SMART-pool ASK1 siRNA was purchased from Thermo Scientific (Reinach, Switzerland) and Microsynth (Balgach, Switzerland; sequences are available upon request).

## Experiments in primary mouse hepatocytes

Male 10-week-old mice were anesthetized, and the inferior vena cava was cannulated with a 23G needle. The liver was perfused with 25–30 ml of pre-warmed perfusion buffer (1× HBSS, 25 mM HEPES, 0.5 mM EGTA, pH 7.4). Thereafter, the liver was digested by 10–15 ml of pre-warmed digestion buffer [DMEM containing 1 g/l D-glucose and 265 mg/l of $CaCl_2$ (Thermo Fisher Scientific, Waltham, MA, USA, cat # 31885), 15 mM HEPES, 1% antibiotics, and 50 CDU/ml of collagenase IV (Sigma cat. # C5138)]. The liver was removed and the hepatocytes released into 10 ml of ice-cold isolation/plating media (DMEM containing 1 g/l D-glucose, 10% FBS, 1% antibiotics) by shaking. Cells were suspended, filtered through a 100-μm cell strainer, and centrifuged at $50 \times g$ at RT for 2 min. The pellet was washed twice with 20 ml of ice-cold isolation/plating media, suspended in 5 ml of ice-cold 1× HBSS, and added to 5 ml of a Percoll gradient solution [4.5 ml of Percoll (Sigma cat. # P1644) and 0.5 ml of 10× HBSS]. Following centrifugation for 10 min at $50 \times g$ at RT, the pellet was washed in 10 ml of ice-cold isolation/plating media. The pellet was suspended in 10 ml of pre-warmed isolation/plating media, and primary hepatocytes were plated on black 96-well plates with clear bottoms at $3.5 \times 10^4$ cells/well. After 6 h, media was changed to serum-free media and cells were treated with or without 0.4 mM BSA, 0.4 mM palmitate, 10 nM bafilomycin A1, or 1 mM rapamycin. After 24 h, cells were fixed with 4% formaldehyde, blocked, and incubated o/n with primary LC3A/B antibody (#12741, Cell Signaling Technology). Subsequently, cells were incubated with a Cy3 anti-rabbit secondary antibody. Fluorescence images were taken, and lipid droplets were quantified as mentioned above.

## RNA extraction and quantitative real-time PCR

Total RNA was extracted using RNA extraction kit from Promega according to the manufacturer's protocol. 1.0 μg of total RNA was converted into first-strand cDNA using GoScript Reverse Transcription kit (Takara

**The paper explained**

**Problem**

Non-alcoholic fatty liver disease (NAFLD) is the most common chronic liver disease and is strongly associated with obesity. It may progress to non-alcoholic steatohepatitis (NASH) and liver fibrosis. Triggering factors in the development of NAFLD and liver fibrosis remain largely unknown but activation of autophagy may be beneficial.

**Results**

Herein, we identify apoptosis signal-regulating kinase 1 (ASK1) as a suppressor of NASH and fibrosis formation. Specifically, liver-specific ASK1 deletion in mice aggravated high-fat diet and age-induced hepatic steatosis, inflammation, and fibrosis. In contrast, liver-specific ASK1 overexpression protected mice from the development of high-fat diet-induced hepatic steatosis and carbon tetrachloride-induced fibrosis. Mechanistically, ASK1 depletion blunts autophagy, thereby enhancing lipid droplet accumulation and the formation of liver fibrosis.

**Impact**

ASK1 may prevent the development of obesity-associated hepatic steatosis and liver fibrosis through induction of autophagy.

or Promega, Switzerland). The following primers were used: *Acox1*, Mm00443579_m1; *Cpt1α*, Mm00550438_m1; *Pparα*, Mm00627559_m1; *Mcp-1*, Mm00441242_m1; *Tnf-α*, Mm00443258_m1; *Il-6*, Mm00446190_m1; *F4/8*, Mm00802529_m1; *Plin*, Mm00558672_m1; *Fas*, Mm00662319_m1; *Acc1*, Mm01304257_m1; *Srebp1*, Mm00550338_m1; *Hsl*, MM00495 359_m1; *Il-1β*, Mm00434228_m1; *Col1a1*, Mm00801666_g1; *Tgfβ1*, MM01178820_m1 (Applied Biosystems, Foster City, CA, USA). For Vldlr, real-time PCR quantification was performed using fast SYBR Green master mix and gene-specific primer sets (sequences are available upon request). Expression of the respective genes was normalized to 18S rRNA (Applied Biosystems) as an internal control.

**Western blot analysis**

Tissue samples and cultured cells were lysed in RIPA buffer and centrifuged at 24,000 *g* for 10 min, and protein concentrations were measured. Equal amounts of proteins were resolved by SDS–PAGE (8–15%) gel and transferred onto a nitrocellulose membrane as previously described (Wueest *et al*, 2010; Challa *et al*, 2012; Item *et al*, 2017). The following primary antibodies were used: anti-ASK1 (ab45178, Abcam, Cambridge, United Kingdom; diluted 1:1,000), anti-phospho-ASK1 (Thr 845, sc-109911; diluted 1:2,000), anti-phospho-JNK (G-7; sc-6254; diluted 1:2,000), and anti-COL1A1 (sc-293182; diluted 1:200) were obtained from Santa Cruz Biotechnology, Heidelberg, Germany. Anti-BCN1 (NB110-87318SS; Novus, Abingdon, United Kingdom; diluted 1:1,000), anti-phospho-p38 MAPK (Thr180/Tyr182, 9211; diluted 1:1,000), anti-p38 MAPK (9212; diluted 1:1,000), anti-JNK2 (56G8; 9258; diluted 1:1,000), anti-ATG12 (D88H11; 4180; diluted 1:1,000), anti-phospho-BCN 1 (Ser15; 3825; diluted 1:1,000), anti-LC3A/B (D3U4C; 12741; diluted 1:1,000), anti-SQSTM1/p62 (5114; diluted 1:1,000), and anti-HSP90 (4877; diluted 1:1,000) were obtained from Cell Signaling Technology. Anti-Actin (MAB1501, Merck Millipore, Darmstadt, Germany; diluted 1:5000), anti-GAPDH (10494-1-AP, Proteintech, Rosemont, IL, USA; diluted 1:1000). Thereafter, membranes were incubated with corresponding secondary HRP–conjugated antibodies. Actin, HSP90 and GAPDH were used as a loading control. The proteins were developed and quantified with Aida Image analyzer version (version 3.52).

**Data analysis**

All data are expressed as mean ± SEM. The significance was determined using a two-tailed, unpaired Student's *t*-test, Spearman correlation, one-way ANOVA with Newman–Keuls correction for multiple group comparisons, or two-way ANOVA with Bonferroni multiple comparisons. Statistical tests were calculated using the GraphPad Prism 5.0 (GraphPad Software, San Diego, CA, USA). *P*-values < 0.05 were considered significant. Exact *P*-values are listed in Appendix Table S4. Power calculation analysis was not performed. Sample size was determined based on previous experiments performed in our laboratory. The evaluator was blinded to the identity of a specific sample as far as the nature of the experiment allowed it.

# Data availability

The datasets supporting the findings of this study are available from the authors on request.

**Expanded View** for this article is available online.

## Acknowledgements

This work was supported by grants from the Swiss National Science Foundation (#310030-160129 and #310030-179344 to DK). We would like to greatly acknowledge Myrtha Arnold (ETH Zurich) for assistance regarding plasma determination of ALT, AST, FFA, and TG, and Heidi Seiler (University Hospital Zurich) for help regarding histology.

## Author contributions

TDC designed and performed experiments, analyzed data, and wrote the manuscript. SW designed and performed experiments and wrote the manuscript. FCL, MD, SM, and MBo performed experiments. CW designed experiments. MBl provided human samples and designed experiments. DK designed experiments, analyzed data, and wrote the manuscript. All authors reviewed and commented on the manuscript.

## Conflict of interest

The authors declare that they have no conflict of interest.

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
