## [Review Process File · EMBO Molecular Medicine]

Liver ASK1 protects from non-alcoholic fatty liver disease and fibrosis

Tenagne D. Challa, Stephan Wueest, Fabrizio C. Lucchini, Mara Dedual, Salvatore Modica, Marcela Borsigova, Christian Wolfrum, Matthias Blüher, Daniel Konrad

Review timeline:	Submission date:	25 November 2018
	Editorial Decision:	11 January 2019
	Revision received:	23 March 2019
	Editorial Decision:	12 April 2019
	Revision received:	30 April 2019
	Accepted:	3 May 2019

Editor: Céline Carret

Transaction Report:

1st Editorial Decision

11 January 2019

Thank you for the submission of your manuscript to EMBO Molecular Medicine. We have now heard back from the three referees whom we asked to evaluate your manuscript.

You will see from the comments pasted below that the 3 reports are consistent, fair and constructive. The referees' comments are overlapping and recommend concentrating on providing more conclusive functional data on 1) autophagy (causative effect), 2) lipophagy, 3) fibrosis and on addressing the discrepancies observed in literature.

We would therefore welcome the submission of a revised version within three months for further consideration and would like to encourage you to address all the criticisms raised as suggested to improve conclusiveness and clarity. Please note that EMBO Molecular Medicine strongly supports a single round of revision and that, as acceptance or rejection of the manuscript will depend on another round of review, your responses should be as complete as possible.

I look forward to receiving your revised manuscript.

***** Reviewer's comments *****

Referee #2 (Comments on Novelty/Model System for Author):

Very good concept, strong rationale but with novel and appropriate animal model. Matched by in vitro data. Experimental design and general data quality good quality. However, measures of autophagy for sure, and fibrosis possibly, could be improved.

Referee #2 (Remarks for Author):

Authors investigated the role of ASK1 in mitigating NASH and liver fibrosis, and related its effects to autophagy rates in liver. The study was composed of excellent animal work design, and investigated both loss- and gain-of function in whole body or tissue specific set-up. As such, it is certainly a study worthy of great credit. Nevertheless, prior to this being published some aspects of the experimental data have to improve. In particular, in my opinion the demonstration of autophagosomes by imaging should be more robust. The following minor manuscript and experimental changes are recommended and presented in a figure by figure format.

Fig2 -

- B. quantitate effect of siRNA (preferably from more than n=3)
- D. show both BSA and palm in graph; it is not clear if quantification shown here was done on basal level, or with palmitate?
- E. LC3B puncta imaging could be more convincing - can the authors corroborate whether this pattern matches conventional size/morphology of autophagosomes? The individual cell response is also quite heterogeneous. Please comment.

Since higher LC3-II levels (western) or puncta (images) could mean more autophagy flux, experiments with lysosomal blocker like bafilomycin A1 or Chloroquine, to show LC3-II accumulation rate should be done on siASK1 cells.

If possible, transmission electron microscope images of siCon versus siASK1 cells would be a nice addition.

Fig 3 -

Examination of collagen at the mRNA level is very weak to infer fibrosis, especially only one collagen isoform was tested here. To make analysis of collagen more robust it may be useful to add scanning electron microscopy if possible, or perhaps H&E with Massons Trichrome stain in tissue sections.

Fig4 -

Again, in D the size/appearance of autophagosomes appears unusual to me and I would like the authors to discuss this and validate their interpretation.

JNK phosphorylates bcl2, which disrupts bcl2-beclin1 interaction (co-immunoprecipitation experiment with beclin1 and bcl2 is desired), and phosphorylation site for Beclin1 presented in this study is also relevant to mTOR activity. To show direct interaction between JNK and autophagy is it possible to inhibit JNK signaling and then test changes in autophagy?

In reading the text I assumed the authors would propose and test if lipophagy was a critical mechanistic component of the observations made here. Rather, the conclusions are made based on individual analysis of lipid changes and general autophagy markers. A more detailed investigation of lipophagy is warranted.

The role of autophagy in NASH and hepatic fibrosis has been established for several years. Could the authors comment on whether translation of these observations to therapies has advanced; that is are any therapeutic approaches targeting autophagy being tested for NASH/fibrosis? If yes, fully update. If not, speculate on how this could be achieved.

Spelling page 16 change "numbers of LC3-II punctuates" to puncta.

Referee #3 (Comments on Novelty/Model System for Author):

The current work reports an unexpected protective action of hepatic ASK1 against NASH. In contrast to these observations, liver ASK1 was previously reported to cause NASH. The molecular mechanism to explain the underlying discrepancy is unknown. Importantly, the study lacks convincing causal-effectual data to support the notion that autophagy mediates the ASK1 action. The mouse phenotypes are modest. Overall, the findings are interesting and potentially important, but are premature due to lack of convincing in depth mechanistic data.

Referee #3 (Remarks for Author):

Challa TD et al reported that hepatocyte-specific deletion of ASK1 augments HFD-induced liver steatosis, inflammation and fibrosis, whereas liver-specific overexpression of ASK1 has the opposite effects. The authors provided some data suggesting that autophagy likely mediates these effects. The study was well designed and performed, and the manuscript was well written. The data are consistent and suggest that hepatic ASK1 likely plays a modest action against NASH, although the phenotypes of the mutant mice are mild. Notably, previous reports observed that aberrant activation of hepatic ASK1 is a causal factor, in contrast to the current results, for NASH. The molecular mechanisms explaining these contradictory conclusions remain unknown. Moreover, the contribution of hepatic autophagy to the observed phenotypes in ASK1-deficient or -overexpressing mice is inadequately investigated. The autophagy data are not convincing, and are insufficient to support the putative ASK1-autophagy-NASH pathway.

1. ASK1 appears to modestly affect some autophagy parameters; however, there is a lack of convincing data to test the notion that these changes mediate liver steatosis, inflammation, and fibrosis in the mutant mice.
2. Given that ASK1-overexpressing mice are protected from CCl₄-induced liver fibrosis, the contributions of lipophagy to HFD-induced liver inflammation and fibrosis in the mutant mice remain unclear.
3. Fig. 3F and G: The Sirius red data are not convincing, and the results are insufficient to support the fibrosis conclusions.
4. It is unclear how hepatic ASK1 has the opposing actions in NASH, as observed in the current and previous studies. Additional experiments and/or discussion is needed to address the discrepancy between these studies.

Referee #4 (Remarks for Author):

ASK1 belongs to the MAP3K protein family and is a member of the MAPK signaling pathway. It activates downstream kinases such as JNK and p38 MAPK and a Raf-independent manner. Upstream regulators include stressors such as oxidative stress and endoplasmic reticulum stress. Aberrant ASK1 activation has been linked to several disease states such as cancer, neurodegenerative, but also cardiometabolic disease.

Challa et al. studied the role of ASK1 in the development of non-alcoholic fatty liver disease and liver fibrosis in the context of obesity. Impaired autophagy contributes to the development of hepatic steatosis. Since ASK is a known regulator of autophagy ASK1, the authors hypothesized that ASK1 is involved in the development of steatosis via the regulation of hepatocyte apoptosis. To corroborate this idea the authors took advantage of sophisticated in vitro and in vivo models. Most importantly, they also provide data from human primary liver samples underlining the relevance of their findings for human physiology.

Overall, the paper is written very well and the red line is easy to follow.

Below you find points of critique that should be addressed:

1. Figure 1: Figure 1A is convincing. The point that ASK1 expression relates to autophagy could be further strengthened by measuring additional autophagy genes, e.g. ATG5, 8, or 12. What is the relationship between liver ASK1 expression and BMI and other parameters of liver health (e.g. AST, ALT)? This reviewer is not sure if statistics in Figure 1B is appropriate, at least it's not a very common method. Why not using one-way ANOVA? Please seek advice from a biostatistician and reanalyze when necessary. Figure legend says that Spearman correlation coefficient is given, but this method is not mentioned in the Method part.
2. Figure 2: Would it be possible to quantify the lipid droplet/LC3-II staining?

3. Figure 3B: The only statistically significant difference is between KO and control animals on a HFD? What about differences between diets?
4. Page 9, line 2 states that plasma levels are increased in the transgenic animals and refers to Suppl. Table 2 - here, there is no difference seen.
5. Figure 5/Suppl. Table 3: Liver triglycerides are increased by approx. 25 % with ASK1 inhibitor. FFA levels were increased at about the same rate, but TG were increased by more than 10-fold. Is there a possible explanation? Do other tissues play a role here, e.g. WAT? Is ASK1 expression comparable between liver and adipose tissue?
6. Figure 6: Does decreased lipid deposition in liver go along with better glucose tolerance and insulin sensitivity? Was this studied?
7. Supplemental Figure 2/3, metabolic phenotyping of transgenic mice: did the authors also perform insulin tolerance tests? Figure S3: Are data on liver inflammation available?
8. A crucial point in this study is duration of HFD. All data reported in this study were generated after 20 weeks of HFD and show a harmful effect of liver ASK1 depletion. The authors clearly state that their data are in contrast to published data and cite two other studies claiming that the absence of liver ASK1 is beneficial. However, at the same time they concede that also in their study HFD for 6 weeks had beneficial effects. So they have a set of data agreeing with literature and another set of data contradicting published data, and only the contradicting data are given. I recommend displaying the own data generated after 6 weeks of HFD (e.g. in the Supplement) and more detailed discussion.

Minor:

Page 10, line 2: typo in collagen

1st Revision - authors' response

23 March 2019

Responses to Referee #2

We thank the Referee for his/her constructive comments and suggestions and we are delighted to learn that he/she found our concept very good and our animal work design excellent.

Concerns:

Fig2 -

B. quantitate effect of siRNA (preferably from more than n=3)

We now quantified the effect of siRNA on ASK1 protein levels (n=6) and added a graph to the revised manuscript (Supplemental Fig 1B).

D. show both BSA and palm in graph; it is not clear if quantification shown here was done on basal level, or with palmitate?

In the submitted manuscript, LC3-II was quantified from bands of the displayed blot (i.e. n=3 (basal and palmitate) for siCtrl and siAsk1). We now repeated such experiment (see Western blot below) and show the quantification of the palmitate bands (n=4 for siCtrl and siAsk1) in Fig. 2D of the revised manuscript.

E. LC3B puncta imaging could be more convincing - can the authors corroborate whether this pattern matches conventional size/morphology of autophagosomes? The individual cell response is also quite heterogeneous. Please comment.

Fig. 2E was intended to support findings of Fig. 2D, i.e. that ASK1 knockdown leads to LC3B accumulation suggesting inhibition of autophagy. From that experiment we cannot corroborate whether this pattern matches conventional size of autophagosomes. We now repeated this experiment with similar results (see below). This experiment suggests increased co-localization of LC3 and lipid droplets in ASK1-depleted hepatocytes, indicating increased lipophagy (see also below). We agree that the individual cell response is heterogeneous and added such statement to the revised manuscript (*Result* section, 2nd paragraph). Nevertheless, we do believe that the overall picture is quite convincing.

Since higher LC3-II levels (western) or puncta (images) could mean more autophagy flux, experiments with lysosomal blocker like bafilomycin A1 or chloroquine, to show LC3-II accumulation rate should be done on siASK1 cells.

As suggested, we performed experiments using the lysosomal blocker bafilomycin A1. Confirming previous experiments, ASK1 knockdown increased LC3 co-localization with lipid droplets in BSA treated cells. Moreover, inhibition of autophagy using bafilomycin A1 significantly increased LC3 accumulation in control hepatocytes as previously reported (Pi H et al., *Autophagy* 2015; 11(7): 1037-51). Importantly, such effect was blunted in ASK1-depleted cells (siCtrl 227±41% vs. siASK1 49±9%, $p < 0.01$), indicating that blunted autophagy contributes to increased LC3II accumulation in siASK1 treated HepG2 cells. These data were added to Fig. 2F of the revised manuscript.

If possible, transmission electron microscope images of siCon versus siASK1 cells would be a nice addition.

We actually planned to do such experiments forehand, however experiments were too expensive and too cumbersome so that we had to drop this idea.

Fig 3 -

Examination of collagen at the mRNA level is very weak to infer fibrosis, especially only one collagen isoform was tested here. To make analysis of collagen more robust it may be useful to add scanning electron microscopy if possible, or perhaps H&E with Massons Trichrome stain in tissue sections.

We agree that mRNA level alone is not sufficient. Therefore, we also stained liver sections with Sirius Red and found significantly increased levels in liver-specific ASK1 knockout mice fed a HFD for 20 weeks (Fig. 3F and 3G) or fed a chow diet for 15 months (Fig. 3K). In agreement, Sirius Red staining was significantly reduced in liver-specific ASK1 overexpressing mice fed a HFD for 20 weeks (Fig. 6D). Of note, Sirius Red may be a superior staining for liver fibrosis compared to Massons Trichrome (Huang Y et al., *Liver Int.* 2013, 33(8):1249-56). In addition, immunohistochemical analysis revealed upregulated alpha-smooth muscle actin (α -SMA) protein

levels in the liver of HFD-fed ASK1^{Δhep} mice (Fig. 3H). We now also determined hepatic protein levels of collagen 1A1 (COL1A1) in liver-specific ASK1 knockout mice fed a HFD for 20 weeks. As shown in Fig. 3I of the revised manuscript, COL1A1 protein was significantly increased in knockout mice, further indicating elevated fibrosis in the latter. Altogether, four different methods used indicated increased liver fibrosis in liver-specific ASK1 knockout mice. We are therefore confident to conclude that depletion of ASK1 increases fibrosis in mice fed a HFD for 20 weeks.

Fig4 -

Again, in D the size/appearance of autophagosomes appears unusual to me and I would like the authors to discuss this and validate their interpretation.

As eluded above we cannot correlate LC3 puncta with size of autophagosomes. While increased LC3 puncta suggest elevated accumulation of this autophagy marker, it does not represent the actual size/appearance of autophagosomes. To validate our findings, we repeated the experiment in HepG2 cells (see above) with similar results (new Fig. 2E). As mentioned above, we added a statement regarding the heterogeneous cell response to the revised manuscript (*Result* section, 2nd paragraph).

JNK phosphorylates bcl2, which disrupts bcl2-beclin1 interaction (co-immunoprecipitation experiment with beclin1 and bcl2 is desired), and phosphorylation site for Beclin1 presented in this study is also relevant to mTOR activity. To show direct interaction between JNK and autophagy is it possible to inhibit JNK signaling and then test changes in autophagy?

It has been previously shown that JNK inhibition blunts autophagy in HepG2 cells (Zhang et al., *PLoS One* 2016; 11(1): e0147405; Wang et al., *Food Chem Toxicol* 2018; 116: 40-50), indicating a direct interaction between JNK and autophagy. Such fact was added to the *Discussion* of the revised manuscript.

In reading the text I assumed the authors would propose and test if lipophagy was a critical mechanistic component of the observations made here. Rather, the conclusions are made based on individual analysis of lipid changes and general autophagy markers. A more detailed investigation of lipophagy is warranted.

We now quantified LC3 colocalization with lipid droplets to assess lipophagy (Singh R et al., *Nature* 2009; 458(7242): 1131-5). As shown in Fig. 2F of the revised manuscript, ASK1 depletion increased LC3 colocalization with lipid droplets, indicating blunted lipophagy. To analyse whether reduced lipophagy may be a mechanistic component, experiments with the lysosomal blocker bafilomycin A1 were performed. Bafilomycin treatment significantly increased LC3 accumulation in control hepatocytes but not in ASK1-depleted cells (Fig. 2F of the revised manuscript). Moreover, we now also determined lipid accumulation after treatment with bafilomycin. In agreement to findings presented in the submitted manuscript (Fig. 2C), ASK1 depletion significantly increased lipid droplet accumulation in hepatocytes treated with or without palmitate compared to cells transfected with non-targeting control siRNA (siCtrl). Importantly, treatment with bafilomycin increased lipid droplet accumulation in siCtrl cells as expected, whereas such effect was blunted in siASK1 treated cells. Hence, blocked lipophagy may be critically involved in the effect of ASK1 on hepatic lipid accumulation. These data are now added to Fig. 2G.

The role of autophagy in NASH and hepatic fibrosis has been established for several years. Could the authors comment on whether translation of these observations to therapies has advanced; that is are any therapeutic approaches targeting autophagy being tested for NASH/fibrosis? If yes, fully update. If not, speculate on how this could be achieved.

As recently reviewed (Allaire M et al., *J Hepatol* 2019, epub ahead of print), molecules targeting autophagy revealed beneficial effects on NASH both in animal and clinical studies. However, tested drugs may have affected other pathways than autophagy and therefore suffered from specificity towards the latter. Therefore, identification of more selective and hepatocyte-specific activators of autophagy to tackle NASH is warranted. Such statement is now added to the *Discussion* of the revised manuscript.

Spelling page 16 change "numbers of LC3-II punctuates" to puncta.

We have changed punctuates to puncta as requested.

Responses to Referee #3

We thank the Referee for his/her encouraging comments and we are grateful that he/she found our findings interesting and potentially important.

Comments:

1. ASK1 appears to modestly affect some autophagy parameters; however, there is a lack of convincing data to test the notion that these changes mediate liver steatosis, inflammation, and fibrosis in the mutant mice.

We now analysed the impact of autophagy on lipid accumulation in cultured hepatocytes using the lysosomal blocker bafilomycin A1 (please see also our answer to *Referee 2* above). In agreement to findings presented in the submitted manuscript (Fig. 2C), ASK1 depletion significantly increased lipid droplet accumulation in HepG2 cells treated with or without palmitate. Importantly, treatment with bafilomycin significantly increased lipid droplet accumulation in siCtrl cells, whereas such effect was blunted in siASK1 treated cells. Such data was now added to Fig. 2G of the revised manuscript. In agreement with these findings, bafilomycin blunted the effect of rapamycin on the clearance of palmitate-induced lipid droplet accumulation in hepatocytes isolated from control but not liver-specific ASK1 knockout mice (Fig. 4C), further indicating that reduced autophagy in ASK1-depleted hepatocytes contributes to elevated lipid accumulation.

While we have shown a protective effect of liver-specific ASK1 on liver fibrosis (Fig. 6), we have not directly assessed the role of impaired autophagy in the development of the latter. However, it has been suggested that defective autophagy is involved in the development of liver inflammation and fibrosis (Inokuchi-Shimizu et al., *J Clin Invest* 2014, 124: 3566-78; Lodder et al., *Autophagy* 2015, 11: 1280-92). In the *Discussion* section, we state accordingly that ASK1 depletion increases hepatic lipid droplet accumulation and/or liver fibrosis *potentially* via blocking autophagy function.

2. Given that ASK1-overexpressing mice are protected from CCl4-induced liver fibrosis, the contributions of lipophagy to HFD-induced liver inflammation and fibrosis in the mutant mice remain unclear.

We now determined LC3 colocalization with lipid droplets to assess lipophagy (Singh R et al., *Nature* 2009; 458(7242): 1131-5). As shown in Fig. 2F of the revised manuscript ASK1 depletion in hepatocytes increased LC3 colocalization with lipid droplets, indicating that ASK1 impacts on lipophagy (please see also our answer to *Referee 2* above). Moreover, experiments using the autophagy inhibitor bafilomycin revealed that autophagy is critically involved in the effects mediated by ASK1. Nevertheless, the contribution of lipophagy, which is a selective form of autophagy, cannot clearly be unravelled.

3. Fig. 3F and G: The Sirius red data are not convincing, and the results are insufficient to support the fibrosis conclusions.

Besides assessing Sirius Red staining in liver sections, we also performed immunohistochemical analysis of alpha-smooth muscle actin (α -SMA) in the liver revealing upregulated protein levels in HFD-fed ASK1^{Δhep} mice (Fig. 3H). Additionally, mRNA expression of *Col1A1* and *Tgfb* were significantly increased in HFD-fed knockout mice (Fig. 3E), further indicating increased fibrosis in the latter. Moreover, we now determined hepatic protein levels of collagen 1A1 (COL1A1) in liver-specific ASK1 knockout mice fed a HFD for 20 weeks. As shown in Fig. 3I of the revised manuscript, COL1A1 protein was significantly increased in knockout mice. Altogether, all four different methods used indicated increased liver fibrosis in liver-specific ASK1 knockout mice. We are therefore confident to conclude that depletion of ASK1 increases fibrosis in mice fed a HFD for 20 weeks.

4. It is unclear how hepatic ASK1 has the opposing actions in NASH, as observed in the current and

previous studies. Additional experiments and/or discussion is needed to address the discrepancy between these studies.

We now provide more detailed information on the phenotype observed in ASK1^{Δhep} and control mice after 6 weeks of HFD. Of note, elevated liver lipid accumulation in ASK1^{Δhep} mice was observed only after 20 but not after 6 weeks of HFD feeding. In fact, liver inflammation (i.e. lower Tnf- α expression) was reduced and glucose tolerance improved when compared to control littermates after 6 weeks of HFD. These data were now added to the manuscript (Supplemental Fig. 5) and we expanded our discussion on this point. Consequently, ASK1 depletion may have a beneficial effect in an earlier phase of high fat feeding, whereas it negatively affects liver lipid metabolism and promotes NASH after prolonged high fat diet intake. We now expanded the *Discussion* to address the discrepancy between the studies.

Responses to Referee #4

We thank the Referee for his/her insightful comments and we are happy to learn that he/she found our *in vitro* and *in vivo* models sophisticated and our paper to be written very well.

Points of critique:

1. Figure 1: Figure 1A is convincing. The point that ASK1 expression relates to autophagy could be further strengthened by measuring additional autophagy genes, e.g. ATG5, 8, or 12.

As suggested, we measured mRNA expression of the autophagy genes *ATG5*, *ATG8* and *ATG12* and correlated each of them with ASK1 expression (results for *ATG5* and *ATG12* are now presented in new Figs. 1E and 1G; please find results for *ATG12* presented below). As demonstrated, expression of all three genes correlated significantly with ASK1 expression.

$$r=0.33; p=0.0136$$

What is the relationship between liver ASK1 expression and BMI and other parameters of liver health (e.g. AST, ALT)? This reviewer is not sure if statistics in Figure 1B is appropriate, at least it's not a very common method. Why not using one-way ANOVA? Please seek advice from a biostatistician and reanalyze when necessary. Figure legend says that Spearman correlation coefficient is given, but this method is not mentioned in the Method part.

As suggested, we correlated hepatic ASK1 mRNA expression with BMI (new Fig. 1A) or ALAT (new Fig. 1B). We report a negative correlation of BMI as well as ALAT with hepatic ASK1 expression, further supporting a positive role of hepatic ASK1 activation in metabolism.

We agree with the Referee that correlation analysis for NASH scores and ASK1 expression was not appropriate and now performed ANOVA analysis (new Fig. 1D). In addition, use of Spearman correlation is now mentioned in *Materials and Methods* of the manuscript.

2. Figure 2: Would it be possible to quantify the lipid droplet/LC3-II staining?

We now quantified LC3 colocalization with lipid droplets in HepG2 cells (please see also our answer to *Referee 2* above). ASK1 knockdown increased LC3 colocalization with lipid droplets compared to siCtrl treated cells, indicating blunted lipophagy (Singh R et al., *Nature* 2009;458(7242):1131-5). Moreover, inhibition of autophagy using bafilomycin A1 significantly increased LC3 accumulation in hepatocytes as previously reported (Pi H et al., *Autophagy* 2015; 11(7): 1037-51). Importantly, such effect was significantly blunted in ASK1-depleted cells (siCtrl 227±41% vs. siASK1 49±9%, $p<0.01$), indicating that blunted autophagy contributes to increased LC3II accumulation in siASK1 treated HepG2 cells. These data were added to Fig. 2F.

3. *Figure 3B: The only statistically significant difference is between KO and control animals on a HFD? What about differences between diets?*

For both genotypes, differences between diets are significantly different. Such information is now shown in Fig. 3B.

4. *Page 9, line 2 states that plasma levels are increased in the transgenic animals and refers to Suppl. Table 2 - here, there is no difference seen.*

Although circulating insulin levels were 50% higher in HFD-fed knockout compared to control mice, such difference was not statistically significant (ASK1^{F/F} 1.0±0.2 ng/ml vs. ASK1^{Δhep} 1.5±0.2 ng/ml; $p=0.18$). We now adapted such statement accordingly.

5. *Figure 5/Suppl. Table 3: Liver triglycerides are increased by approx. 25 % with ASK1 inhibitor. FFA levels were increased at about the same rate, but TG were increased by more than 10-fold. Is there a possible explanation? Do other tissues play a role here, e.g. WAT? Is ASK1 expression comparable between liver and adipose tissue?*

We thank the Referee for this excellent remark. Indeed, we have evidence that ASK1 expression in WAT plays a role in glucose and lipid metabolism. A manuscript relating to this finding is currently under review elsewhere.

6. *Figure 6: Does decreased lipid deposition in liver go along with better glucose tolerance and insulin sensitivity? Was this studied?*

We performed glucose tolerance tests in liver-specific ASK1 overexpressing (n=8) and control (n=6) mice and found improved glucose tolerance in the former (AUC ASK1^{F/F} 2267±195 mmol/l*min vs. ASK1^{+hep} 1615±116 mmol/l*min; $p<0.05$). These data were now added to the *Result* section of the revised manuscript.

7. *Supplemental Figure 2/3, metabolic phenotyping of transgenic mice: did the authors also perform insulin tolerance tests? Figure S3: Are data on liver inflammation available?*

Insulin tolerance test were performed. However, we did not find significant differences between the genotypes neither in mice fed a HFD for 20 weeks nor in 15 months old chow-fed mice. We had access to liver-RNA of a limited number of 15 months old chow-fed mice (n=4 for ASK1^{F/F} and n=7 for ASK1^{Δhep}). mRNA analysis in these livers showed a non-significant increase in *Il-6* and *Il-1β* expression in knockout compared to control mice (see below). This is in agreement with data found in 6 months old chow-fed mice, where *Il-6* mRNA was significantly increased (1.0±0.2 in ASK1^{F/F} vs. 2.4±0.7 in ASK1^{Δhep}, $p<0.05$). As the analysis in 15 months-old chow-fed may be underpowered, we decided not to include it in the revised manuscript.

Relative mRNA expression of respective genes in liver of 15 months-old chow-fed mice (ASK1^{F/F} n=4; ASK1^{hep} n=7). All values are expressed as mean \pm SEM.

8. A crucial point in this study is duration of HFD. All data reported in this study were generated after 20 weeks of HFD and show a harmful effect of liver ASK1 depletion. The authors clearly state that their data are in contrast to published data and cite two other studies claiming that the absence of liver ASK1 is beneficial. However, at the same time they concede that also in their study HFD for 6 weeks had beneficial effects. So they have a set of data agreeing with literature and another set of data contradicting published data, and only the contradicting data are given. I recommend displaying the own data generated after 6 weeks of HFD (e.g. in the Supplement) and more detailed discussion.

Data received in mice after 6 weeks of HFD were now added to the revised manuscript (Supplemental Fig. 5). Moreover, we now expanded the *Discussion* to address the discrepancy to published data.

Minor:

Page 10, line 2: typo in collagen.

We apologize for this typo and corrected it as requested.

2nd Editorial Decision

12 April 2019

Thank you for the submission of your revised manuscript to EMBO Molecular Medicine. We have now received the enclosed reports from the referees that were asked to re-assess it. As you will see the reviewers are now globally supportive and I am pleased to inform you that we will be able to accept your manuscript pending minor editorial amendments.

I look forward to reading a new revised version of your manuscript as soon as possible.

***** Reviewer's comments *****

Referee #2 (Comments on Novelty/Model System for Author):

In my opinion the immunofluorescent detection of LC3 is very unusual in appearance - see figure 2 and especially figure 4. If you have an editorial board member who is expert on autophagy perhaps they could give a second opinion.

Saying that "experiments were too expensive and too cumbersome" to do is disappointing

Referee #2 (Remarks for Author):

The authors have responded to many of the comments raised in initial review.

Yet analysis of autophagy using microscopy remains to be resolved and the appearance of LC3 puncta are not typical of autophagosome size/punctate structure.
The heterogenous response is normal, and that is not a problem.
I also cannot see increased co-localization of LC3 and lipid droplets in ASK1-depleted hepatocytes from the figure shown in response.
Also, some images in the revised version / response to referees are blurry and perhaps out of focus (especially high magnification).

Referee #3 (Remarks for Author):

The authors have addressed concerns with new data. The manuscript has been improved, and I have no further questions.

Referee #4 (Remarks for Author):

My points of critique were adequately addressed.

2nd Revision - authors' response

30 April 2019

We are delighted to learn that the reviewers are now globally supportive and that you are able to accept our manuscript pending the final amendments. We now addressed the comments raised by you and referees #2 and #4:

*Provide higher resolution images for figures 2 and 4 immunofluorescence.
Provide better colocalisation experiment, add quantification and critically discuss this finding. We and referee #4 agree with referee #2 that an increased co-localization of LC3 and lipid droplets in ASK-depleted hepatocytes is not visible. Lipid droplets and LC3 puncta are present in parallel in some cells, but they do not co-localize, there is no spatial overlap of green and red staining.*

In the revised manuscript, we now provide better images for Fig. 2E and Fig. 5B (Fig. 4D of previously submitted manuscript – we split Fig. 4 into two figures since we were not able to fit it within one page). In particular, we now provide a zoomed-in image for LC3 and lipid staining of HepG2 cells. In addition, quantification of colocalization of such experiment is now added to the revised manuscript (Appendix Fig. 1F). We agree that lipid droplets and LC3 puncta are present in parallel rather than overlapping. This fact, which is in agreement with recent colocalization experiments performed in hepatocytes (Wang L et al., *Sci Rep* 2017; 7(1): 12307), was added to the *Discussion* of the revised manuscript.

Corresponding Author Name: Daniel Konrad

Journal Submitted to: EMBO Mol Med

Manuscript Number: EMM-2018-10124